# Large Language Models Explore by Latent Distilling

**Yuanhao Zeng** [1 2]  **Ao Lu** [1]  **Lufei Li** [1]  **Zheng Zhang** [1 2]  **Yexin Li** [2]  **Kan Ren** [1]

## Abstract

Generating diverse responses is crucial for test-time scaling of large language models (LLMs), yet standard stochastic sampling mostly yields surface-level lexical variation, limiting semantic exploration. In this paper, we propose Exploratory Sampling (`ESamp`), a decoding approach that explicitly encourages semantic diversity during generation. `ESamp` is motivated by the well-known observation that neural networks tend to make lower-error predictions on inputs similar to those encountered before, and incur higher prediction error on novel ones. Building on this property, we train a lightweight Distiller at test time to predict deep-layer hidden representations of the LLM from its shallow-layer representations to model the LLM's depth-wise representation transitions. During decoding, the Distiller continuously adapts to the mappings induced by the current generation context. `ESamp` uses the prediction error as a novelty signal to reweight candidate token extensions conditioned on the current prefix, thereby biasing decoding toward less-explored semantic patterns. `ESamp` is implemented with an asynchronous training–inference pipeline, with less than 5% worst case overhead (1.2% in the optimized release). Empirical results show that `ESamp` significantly boosts the Pass@k efficiency of reasoning models, showing superior or comparable performance to strong stochastic and heuristic baselines. Notably, `ESamp` achieves robust generalization across mathematics, science, and code generation benchmarks and breaks the trade-off between diversity and coherence in creative writing. Our code has released at: https://github.com/LinesHogan/tLLM.

[1]School of Information Science and Technology, ShanghaiTech University, Shanghai, China [2]State Key Laboratory of General Artificial Intelligence, BIGAI, Beijing, China. Correspondence to: Yexin Li <liyexin@bigai.ai>, Kan Ren <renkan@shanghaitech.edu.cn>.

*Proceedings of the 43rd International Conference on Machine Learning*, Seoul, South Korea. PMLR 306, 2026. Copyright 2026 by the author(s).

## 1. Introduction

Training-free test-time scaling methods have emerged as powerful paradigms for enhancing the reasoning capabilities of large language models (LLMs) (Snell et al., 2024). By generating a batch of candidate solutions and applying selection mechanisms such as reranking (Cobbe et al., 2021), self-verification (Weng et al., 2023), or majority voting (Wang et al., 2023), these approaches consistently outperform greedy decoding.

However, the effectiveness of test-time scaling is fundamentally constrained by the diversity of underlying reasoning strategies present in the candidate set (Dorner et al., 2025; Chen et al., 2025b). If the model produces candidates that are lexically different yet rely on the same core reasoning structure, such as repeating identical logic or systematic failure patterns, subsequent selection mechanisms may be unable to recover correct solutions (Wang et al., 2025).

Unfortunately, naive sampling strategies (Ackley et al., 1985) frequently fall into this trap, as stochastic perturbations at the token level tend to induce surface-level lexical variation without substantially altering the underlying reasoning strategy. As a result, simply increasing the number of sampled candidates often leads to highly redundant solutions, yielding diminishing returns in downstream selection. Attempts to address this limitation include structured search algorithms that explore solution trees (Yao et al., 2023; Kool et al., 2019; Vijayakumar et al., 2018) and heuristic sampling constraints that modify probability distribution (Zhang et al., 2024; Chen et al., 2025a; Holtzman et al., 2020; Minh et al., 2025). However, structured search methods typically incur substantial computational and latency overhead, while heuristic constraints primarily reshape surface distributions and remain limited in their ability to elicit genuinely novel solution strategies (Wang et al., 2025). We argue that effective test-time scaling requires a mechanism that can efficiently encourage novelty in the model's underlying reasoning behavior.

In this paper, we propose **Exploratory Sampling** (`ESamp`), a decoding algorithm that encourages LLM to explore by penalizing tokens that are consist with predictable latent representations during generation, thereby steering generation toward less-explored semantic regions.

Our method is motivated by RND (Burda et al., 2019) and grounded in the observation that neural networks tend to make more accurate predictions on mappings that they have encountered before, while exhibiting larger prediction errors on previously unseen ones. Building on this property, we introduce a lightweight **Latent Distiller** (LD) that is trained online at test time to approximate the mapping from shallow-layer to deep-layer hidden representations within the LLM. As the distiller adapts to the representation mappings induced by the current generation context, familiar mappings yield low prediction error, whereas unfamiliar ones produce higher error. We interpret such high-error mappings as indicators of under-explored semantic or reasoning behaviors.

To incorporate this novelty signal into decoding, we formulate generation as a KL-regularized optimization objective (Peters et al., 2010), where deviations from the base model are softly constrained while encouraging exploration. Within this framework, the model is rewarded for selecting token extensions that lead to under-explored internal representation mappings. This objective admits a closed-form optimal solution: the token distribution of the base model is reweighted by an exponential function of the novelty reward, analogous to entropy-regularized policy optimization (Mudgal et al., 2024).

In practice, we approximate this reweighting by projecting the distiller's prediction error onto the next candidate tokens conditioned on the current prefix, using the resulting scores to bias sampling. This procedure naturally suppresses probability mass on redundant continuations that correspond to familiar representation mappings, while steering generation toward less-explored semantic behaviors. Moreover, because the distiller is updated online across the entire candidate batch, parallel sequences implicitly coordinate to avoid repeating the same underlying reasoning patterns, enabling more effective batch-level exploration.

Implemented via an asynchronous pipeline incurring less than 5% throughput overhead[1] in standard serving scenarios, ESamp significantly enhances diversity without sacrificing generation quality. Unlike heuristic methods that may hamper capabilities in specific domains (e.g., code generation), ESamp demonstrates robust effectiveness across diverse model families and tasks, especially boosting the Pass@k (Chen, 2021) efficiency of reasoning models, outperforming strong stochastic and heuristic baselines

---

[1]The throughput number reported in the main paper was measured with an internal research implementation. Our open-source implementation, released as ESamp in the tLLM framework at https://github.com/LinesHogan/tLLM, obtains higher throughput in the aligned benchmark reported in Appendix D. This open-source version intentionally preserves a general producer–consumer interface for future developer extensions. Further throughput gains are possible if one sacrifices part of this generality and extensibility for task-specific specialization.

In summary, our contributions are:

- We introduce ESamp, a novel decoding method that effectively encourages exploration in LLM generation, particularly with reasoning models that have reflective abilities.

- We present comprehensive experiments across benchmarks, model families and sizes, showing that ESamp outperforms standard stochastic sampling and heuristic methods, exhibiting superior Pass@k scaling efficiency, particularly empowering Pass@k in reasoning models with significantly smaller sampling budgets.

- We implement a high-efficiency asynchronous pipeline that decouples the distiller's training and inference from the main LLM generation. This design ensures that ESamp incurs negligible latency overhead in standard serving scenarios, making it practical for large-scale deployment.

## 2. Related Work

### 2.1. Decoding Strategies to Generate Diverse Responses

**Stochastic Sampling**  To mitigate the degeneration of deterministic decoding, stochastic strategies typically introduce heuristic constraints to truncate the probability distribution. Methods such as Top-$p$ (Holtzman et al., 2020) and its adaptive variants, including Min-$p$ (Minh et al., 2025) and entropy-based sampling (Zhang et al., 2024; Chen et al., 2025a), achieve this by drawing tokens from a restricted candidate pool to inject randomness. While computationally efficient, these approaches primarily produce lexical diversity and often fail to distinguish between meaningful semantic exploration and superficial syntactic variation.

**Structured Search**  In contrast, structured search algorithms, including Diverse Beam Search (Vijayakumar et al., 2018), Stochastic Beam Search (Kool et al., 2019), and Tree of Thoughts (Yao et al., 2023), treat generation as a tree-based exploration problem. These approaches explicitly traverse the solution space to uncover high-quality reasoning trajectories. However, their reliance on multiple explicit branches or backtracking introduces substantial computational overhead, making them impractical for high-throughput generation tasks.

### 2.2. Steering Generation via Logit-Level Control

While heuristic methods such as Contrastive Decoding (Li et al., 2023) explored logit-level modifications, Controlled Decoding (Mudgal et al., 2024) formalized this process as a token-level KL-regularized reinforcement learning problem, showing that optimal steering can be achieved by reweighting logits with a learned value function. Building on this

theoretical framework, subsequent works, such as DeRa (Liu et al., 2024) and OverRIDE (Shi & Pan, 2026), adopt similar formulations for controlled generation.

In this regard, our work is conceptually most similar to OverRIDE (Shi & Pan, 2026), which also introduces online adaptation to suppress redundancy. However, a critical distinction lies in the abstraction level: OverRIDE operates in the vocabulary space, penalizing token repetition, which may fail to capture semantically equivalent sequences expressed with different surface forms. In contrast, `ESamp` utilizes `LD` to estimate redundancy in the continuous representation space. This allows us to penalize recurring semantics rather than specific tokens, achieving robust exploration even when surface forms vary.

## 3. Problem Formulation

We model LLM generation as an MDP $(\mathcal{S}, \mathcal{V}, \pi_\theta)$: at step $t$, the state $s_t = (z_1, \ldots, z_{t-1})$ is the token prefix, $\mathcal{V}$ is the vocabulary, and $\pi_\theta(z_t \mid s_t)$ is the policy induced by the LLM weights $\theta$. Standard decoding selects tokens by maximizing likelihood or applying heuristic truncations (Holtzman et al., 2020), which tends to produce lexically varied but semantically redundant candidates. For tasks requiring reasoning or creativity, effective generation demands exploring the solution space to uncover diverse valid trajectories.

To this end, we consider a parallel batch of $K$ trajectories and introduce a per-step intrinsic reward $r(s_t, z_t)$ that measures the degree to which selecting token $z_t$ steers generation toward semantic regions not yet explored by the batch. Following the KL-regularized policy optimization framework (Peters et al., 2010; Korbak et al., 2022), we optimize:

$$J(\pi) = \mathbb{E}_\pi\big[r(s_t, z_t)\big] - \alpha \, \mathrm{KL}\big(\pi(\cdot \mid s_t) \,\|\, \pi_{\mathrm{ref}}(\cdot \mid s_t)\big), \quad (1)$$

where $\pi_{\mathrm{ref}}$ is the frozen pre-trained LLM and $\alpha > 0$ controls regularization strength. This objective admits a closed-form optimal policy (derivation in Appendix A.1):

$$\pi^*(z \mid s) \propto \pi_{\mathrm{ref}}(z \mid s) \, \exp\!\Big(\tfrac{1}{\alpha} \, r(s, z)\Big). \quad (2)$$

The per-step formulation above is valid when the novelty reward satisfies a *vanishing redundancy* condition: once any trajectory in the batch has explored a semantic region, subsequent visits to that region yield near-zero reward, causing the sequential $Q$-function to reduce to the immediate reward $Q^*(s_t, z_t) \approx r(s_t, z_t)$. We formalize this condition and discuss its practical relaxation in Appendix A.2.

Our goal is to construct an online estimator for $r(s, z)$ that captures how each candidate token redirects generation toward under-explored semantic regions, thereby realizing $\pi^*$ in practice.

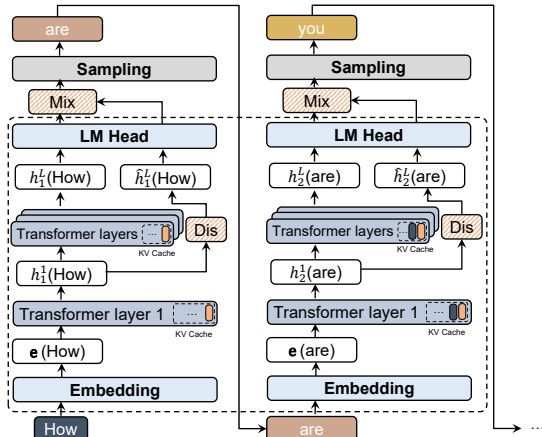

*Figure 1.* The overall framework of our method. `ESamp` intervenes solely during the decode phase. When the first transformer layer outputs hidden states, the Latent Distiller takes them as input to predict the output of the transformer stack's final layer. The predicted hidden states are projected into the vocabulary space via the shared Language Modeling Head to obtain distilled logits. Finally, these logits are mixed with the original LLM logits to serve as the source for sampling. The transformer layer gather information from context, where `LD` get the context-aware representation.

## 4. Methodology

In this section, we propose **Exploratory Sampling** (`ESamp`), a decoding method that promotes semantic exploration. Our approach assesses predictability in the LLM's internal representations to identify recurring semantic patterns in the generated history. Central to our method is the **Latent Distiller** (`LD`), a lightweight module trained online to model the LLM's depth-wise representation transitions from early to late hidden layers, enabling us to discourage semantically redundant generations even when surface forms differ.

### 4.1. Novelty Estimation via Latent Distiller

Standard exploration strategies typically operate in the token space. Rather than operating in token space, we ground exploration in the model's internal representations. By leveraging hidden-layer representations of the generated prefix, we obtain a continuous embedding of the current context that captures its semantic content. Exploration is then guided by encouraging generated contexts to occupy diverse regions of the representation space, rather than merely inducing superficial lexical variation.

Motivated by RND (Burda et al., 2019), we introduce a lightweight Multi-Layer Perceptron (MLP), termed the `LD` $f_\phi$ parameterized by $\phi$, to learn a mapping from shallow-layer to deep-layer hidden representations of the LLM. Specifically, given the hidden representation of the generated prefix at an early layer $h_t^1$, the distiller predicts the

corresponding deep-layer representation $\hat{h}_t^L$:

$$\hat{h}_t^L = f_\phi(h_t^1). \tag{3}$$

The distiller is trained online by minimizing a mean squared error objective over hidden representations encountered during generation. As training proceeds, $f_\phi$ progressively captures representation mappings associated with previously generated contexts. Consequently, low prediction error indicates that the current representation mapping is consistent with past contexts and thus semantically redundant, whereas high prediction error reflects a deviation in representation space, signaling a novel semantic configuration.

## 4.2. Novelty-Driven Generation

Our goal is to quantify how each candidate token $z \in \mathcal{V}$ contributes to extending the prefix toward a more novel semantic region with an action-dependent intrinsic reward $r(s, z)$. We achieve this by mapping both the true deep-layer representation and the distiller-predicted representation into the vocabulary space using the frozen language modeling head $W_{\text{head}}$ of the LLM:

$$\pi_{\text{ref}} = \text{softmax}(W_{\text{head}} h_t^L) \tag{4}$$

$$q_{\text{dist}} = \text{softmax}(W_{\text{head}} \hat{h}_t^L) \tag{5}$$

Here, $q_{\text{dist}}$ represents the probability distribution expected by the LD. We define the intrinsic reward as the log-likelihood ratio: $r(s, z) = \log \pi_{\text{ref}}(z|s) - \log q_{\text{dist}}(z|s)$. Substituting this into the optimal KL-regularized policy derivation in Eq. (2), we obtain the sampling distribution:

$$\pi_{\text{new}}(z|s) \propto \pi_{\text{ref}}(z|s) \cdot \exp\left(\beta \cdot r(s, z)\right)$$
$$\propto \frac{\pi_{\text{ref}}(z|s)^{1+\beta}}{q_{\text{dist}}(z|s)^\beta}; \tag{6}$$

where $\beta = \frac{1}{\alpha}$ is a hyperparameter controlling the exploration intensity. A more detailed derivation is provided in Appendix A.

**Geometric Interpretation.** In the logit space, the operation in Eq. (6) is equivalent to a linear combination of the reference and distilled logits. Let $\text{logit}_{\text{ref}} = W_{\text{head}} h_t^L$ and $\text{logit}_{\text{dist}} = W_{\text{head}} \hat{h}_t^L$, then the new sampling logits are:

$$\text{logit}_{\text{new}} = \text{logit}_{\text{ref}} + \beta(\text{logit}_{\text{ref}} - \text{logit}_{\text{dist}}) \tag{7}$$

Rearranging Eq. (7), we obtain:

$$\Delta \text{logit} = \text{logit}_{\text{new}} - \text{logit}_{\text{ref}} = \beta(\text{logit}_{\text{ref}} - \text{logit}_{\text{dist}})$$
$$= \beta W_{\text{head}}(h_t^L - \hat{h}_t^L) \tag{8}$$

Let $\mathbf{e}_t = h_t^L - \hat{h}_t^L$, which is the **latent error vector** at step $t$, then the logit change for a token $z$ can be written as:

$$\Delta \text{logit}_z = \beta w_z \cdot \mathbf{e}_t = \beta \underbrace{\|w_z\|_2 \cdot \|\mathbf{e}_t\|_2}_{\text{Novelty}} \cdot \underbrace{\cos(w_z, \mathbf{e}_t)}_{\text{Direction}} \tag{9}$$

where $w_z$ denotes the row of $W_{\text{head}}$ corresponding to $z$.

Eq. (9) highlights two critical aspects of our sampling mechanism:

- **Context Novelty:** Euclidean norm of the latent error vector $\|\mathbf{e}_t\|_2$ quantifies the novelty of the current generation context, technically aligned with the prediction error signal in RND. A larger norm indicates that the current context substantially differs from previously explored contexts, thereby automatically signaling stronger exploration.

- **Semantic Direction:** Cosine similarity between $\mathbf{e}_t$ and $w_z$ acts as a selective guide. As $\mathbf{e}_t$ encodes the representation component that the distiller fails to predict, promoting tokens aligned with $\mathbf{e}_t$ directs generation toward semantically distinct trajectories, e.g., novel reasoning, rather than superficial lexical variations.

Consequently, the policy in Eq. 6 discourages redundant semantic responses rather than merely frequent tokens, biasing the generation toward unexplored semantic regions.

## 4.3. Collaborative Exploration via Shared Online Training

A critical advantage of our method emerges in parallel generation settings, where a batch of $K$ sequences is generated simultaneously. Since the LD $f_\phi$ is updated online using hidden representations from *all* sequences, it serves as a shared communication channel that coordinates exploration.

The mechanism functions as an implicit "first-come, first-served" scheduler. This is conceptually similar to centralized exploration signals used in multi-agent reinforcement learning to coordinate agents and avoid redundant coverage (Zhang et al., 2022; Jiang et al., 2024; Chaslot et al., 2008). When sequence A explores a latent pattern, the Distiller learns the corresponding mapping, causing a high match between $q_{\text{dist}}$ and $\pi_{\text{ref}}$ for that region. By Eq. (6), this suppresses the probability of subsequent sequences revisiting the same semantic mode. This effectively *suppresses* the likelihood of re-visiting semantic modes already claimed by other parallel workers, forcing the collective generation to diverge and cover the semantic space efficiently without explicit heuristic penalties. This coordination directly reflects the reward structure described in Section 3: once a trajectory explores a semantic region, the Distiller learns the corresponding mapping, causing the novelty reward for that region to vanish for all subsequent trajectories. Per-step suppression thus compounds into trajectory-level divergence.

**Algorithm Procedure.** The complete execution flow is summarized in Algorithm 1. The process consists of two parallel streams: the *Distiller Prediction Stream* and the

**Algorithm 1** Exploration Sampling with Online Latent Distilling (Decode Step)

---

**Require:** LLM components ($W_{\text{embed}}$, Layers, $W_{\text{head}}$), Distiller $f_\phi$, Current batch inputs $s_1$
**Require:** Hyperparameters: Exploration intensity $\beta$, Learning rate $\eta$

1: Initialize $\phi$ randomly
2: **for** step $t = 1$ to $T$ **do**
3:     $x_t \leftarrow \text{LastToken}(s_{t-1})$
4:     **— Phase 1: Forward & Logits Process —**
5:     // Compute LLM 1st layer features
6:     $H_t^1 \leftarrow \text{Layer}_1(W_{\text{embed}}(x_t) \mid x_{<t})$
7:     // Distiller Prediction (Async Stream)
8:     $\hat{H}_t^L \leftarrow f_\phi(H_t^1)$
9:     // Main LLM Execution (Main Stream)
10:     **for** layer $l = 2$ to $L$ **do**
11:       $H_t^l \leftarrow \text{Layer}_l(H_t^{l-1} \mid x_{<t})$     ▷ Heavy Forward
12:     **end for**
13:     // Logits Projection & Fusion
14:     $\text{logits}_{\text{ref}}, \text{logits}_{\text{dist}} \leftarrow W_{\text{head}}(H_t^L, \hat{H}_t^L)$
15:     $\text{logits}_{\text{ref}} \leftarrow \text{Process}(\text{logits}_{\text{ref}})$  ▷ e.g. TopK (optional)
16:     $\text{logits}_{\text{new}} \leftarrow (1 + \beta)\text{logits}_{\text{ref}} - \beta\text{logits}_{\text{dist}}$
17:     **— Phase 2: Online Update —**
18:     // Distiller Training (Async, GPU mainly)
19:     $\mathcal{L} \leftarrow \text{MSE}(H_t^L, \hat{H}_t^L)$     ▷ Batch-averaged loss
20:     $\phi \leftarrow \phi - \eta\nabla_\phi\mathcal{L}$     ▷ Distiller Update
21:     // Sampling & vLLM worker (Main, CPU mainly)
22:     $z_t \sim \text{Softmax}(\text{logits}_{\text{new}})$     ▷ Sample next token
23:     $s_t \leftarrow s_{t-1} + z_t$     ▷ Append next token
24:     worker busy     ▷ vLLM engine process & schedule
25: **end for**

---

*Distiller Training Stream.*

At each time step $t$, we first compute the exploration-biased logits using the current Distiller, which are then used to guide subsequent generation. The Distiller is then trained by minimizing the MSE between its predicted hidden representations and the true representations, aggregated across the batch $B$:

$$\mathcal{L}(\phi) = \frac{1}{|B|} \sum_{i \in B} \|h_{t,i}^L - f_\phi(h_{t,i}^1)\|_2^2 \tag{10}$$

By performing a gradient descent step on parameters $\phi$ at every token or mini-batch of tokens, the Distiller $f_\phi$ remains strictly synchronized with the evolving context of the current generation session.

### 4.4. Asynchronous Implementation

To minimize the computational overhead of the extra projection and backpropagation, we implement an asynchronous pipeline, as illustrated in Figure 2 and Algorithm 1. The asynchronous algorithm portion highlighted in blue in Al-gorithm 1 can be overlapped by the subsequent portion in gray. We decouple the lightweight Distiller operations from the main generation backbone, executing them on a parallel computing stream.

The pipeline overlaps the Distiller's lifecycle with the host LLM's non-critical phases:

1. **Inference overlapping.** The Distiller's forward pass is triggered immediately after the LLM's first layer. While the host LLM executes middle transformer layers, which are often memory-bandwidth bound in practice, the lightweight Distiller computes $\hat{h}_t^L$ in parallel. In typical settings, the distillation logits are ready before the LLM reaches the final layer, enabling logit fusion without stalling.

2. **Hidden training.** To avoid blocking the generation of the current token, the Distiller's training step—i.e., its backward pass and weight update—is deferred to the post-processing interval. This interval, which consists of CPU-bound tasks such as sampling, de-tokenization, and scheduling, often leaves the GPU underutilized, creating an opportunity window for low-impact updates.

By removing the Distiller from the critical path of token generation, our method incurs *negligible* end-to-end overhead in scenarios where such slack exists. Under fully saturated GPU utilization, however, the benefits of overlap may diminish. Implementation details on CUDA stream synchronization and memory management are provided in Appendix B.

## 5. Experiments

In this section, we evaluate the effectiveness of ESamp across a diverse set of reasoning and creative writing tasks. Our experiments aim to address the following research questions: (1) Does ESamp effectively explore at test-time to find valid solutions in complex reasoning tasks? (2) Does ESamp promote semantic diversity rather than just surface-level lexical variation? (3) How do the online distillation and decoding dynamics evolve during generation? and (4) What is the computational overhead of the proposed asynchronous pipeline?

### 5.1. Experimental Setup

**Benchmarks.** We evaluate performance across three distinct domains: **(1) Mathematics:** AIME 2024 and AIME 2025 (Zhang & Math-AI, 2024; 2025), representing challenging competition-level math problems; **(2) Science:** GPQA-Diamond (Rein et al., 2024), a collection of challenging multiple-choice questions in biology, physics, and chemistry. It includes a subset of 198 problems for which two independent domain experts reached consensus on the correct answer, while the majority of non-experts answered

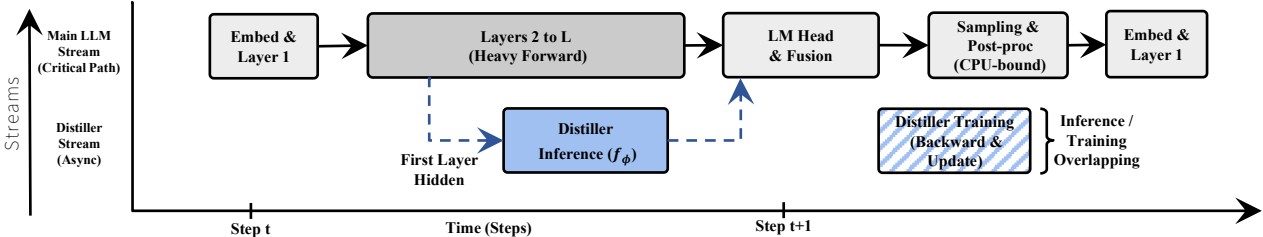

*Figure 2.* A diagram illustrating how the training and inference of the distiller overlap during LLM runtime. The distiller is able to overlap because it eliminates the temporal dependency on the LLM forward pass, thus allowing the distiller to run concurrently with the LLM forward pass.

incorrectly; **(3) Code Generation:** LiveCodeBench v5 (Jain et al., 2024), a holistic benchmark evaluating coding capabilities with 167 problems collected from LeetCode, AtCoder, and Codeforces. **(4) Creative Writing:** Following the protocol in Contrastive Decoding (Li et al., 2023), we use BookCorpus (Zhu et al., 2015) to evaluate story generation. We split stories in the middle, using the last 32 words of the first half as the prompt to generate 512-token continuations. Given that mathematical reasoning tasks often require extensive intermediate steps, we set the maximum context length to 8192 for math benchmarks, while maintaining 4096 for all others. Further details are provided in Appendix E.2.

**Models.** To assess generalizability, we experiment with models of varying capabilities and architectures: general-purpose instruction-tuned models (Qwen2.5-7B/32B-Instruct (Qwen et al., 2025)), general-purpose reasoning models (Qwen3-8B (Yang et al., 2025)) and LLM from other model family (GPT-OSS-20B (OpenAI, 2025)). To prevent excessive reasoning from exhausting the context window, we configured Qwen3 to "no_thinking" mode and GPT-OSS-20B to "medium thinking effort" mode.

**Baselines.** We compare ESamp against three categories of decoding methods: **Stochastic Sampling:** We evaluate vanilla temperature sampling (Holtzman et al., 2020) (temperature=1), Min-$p$ (Minh et al., 2025) ($p_{base}$=0.1), and FIRE (Chen et al., 2025a) which use high temperature at first token and low temperature afterwards. **Structured Search:** We evaluate Tree of Thoughts (Yao et al., 2023) which explores multiple paths by iterative candidate generation and self-evaluation. [2] **Logit-Level Control:** We evaluate Contrastive Decoding (Li et al., 2023) which samples from the difference in logits between a large and small model, and OverRIDE (Shi & Pan, 2026) which trains auxiliary heads at test time to suppress previously generated tokens.

---

[2]We also evaluated Stochastic Beam Search (Kool et al., 2019) and Diverse Beam Search (Vijayakumar et al., 2018). Since beam search inherently lacks diversity in free-form generation, we defer these results to the Appendix E.3.

**Evaluation Metrics.** To evaluate both the quality and diversity of the generated responses, we employ: (1) **Pass@$k$:** The probability of finding at least one correct solution among $k$ samples. (2) **Embedding Similarity (Sim.):** The average pairwise cosine similarity of the generated response embeddings, measuring semantic closeness within generations. (3) **Vendi Score (Friedman & Dieng, 2023):** A spectral metric quantifying the effective number of unique semantic clusters in a batch. (4) **Perplexity (PPL):** A proxy for linguistic coherence, calculated using Qwen2.5-1.5B-Instruct.

**Implementation.** We implement ESamp on top of vLLM (Kwon et al., 2023). The LD consists of a 2-layer MLP trained online, with hidden size of 384 dimensions. Additional details are in Appendix E.2.

## 5.2. Main Results

**ESamp Enables Effective Test-Time Exploration.** Figure 3 presents the Pass@$k$ performance scaling. While baseline methods designed for diversity often show promise at low $k$, they are frequently surpassed by vanilla sampling as $k$ increases, indicating that their exploration strategies may not scale effectively.

In contrast, ESamp demonstrates superior or comparable performance to baselines, particularly excelling with reasoning models. For instruction-tuned models, ESamp consistently ranks among the top decoding strategies. Although certain methods exhibit benchmark-specific advantages, they often fail to generalize. For instance, FIRE extends the performance frontier on AIME24/25 but hampers capabilities on LiveCodeBench v5. In contrast, ESamp shows robust generalization. Even with $\beta$ fixed at 0.25 to prevent hyperparameter cherry-picking, it remains effective in various tasks.

The impact of ESamp is even more pronounced with reasoning models. It not only matches the high-$k$ performance of competing methods with a significantly lower sampling budget but also pushes the performance envelope on several

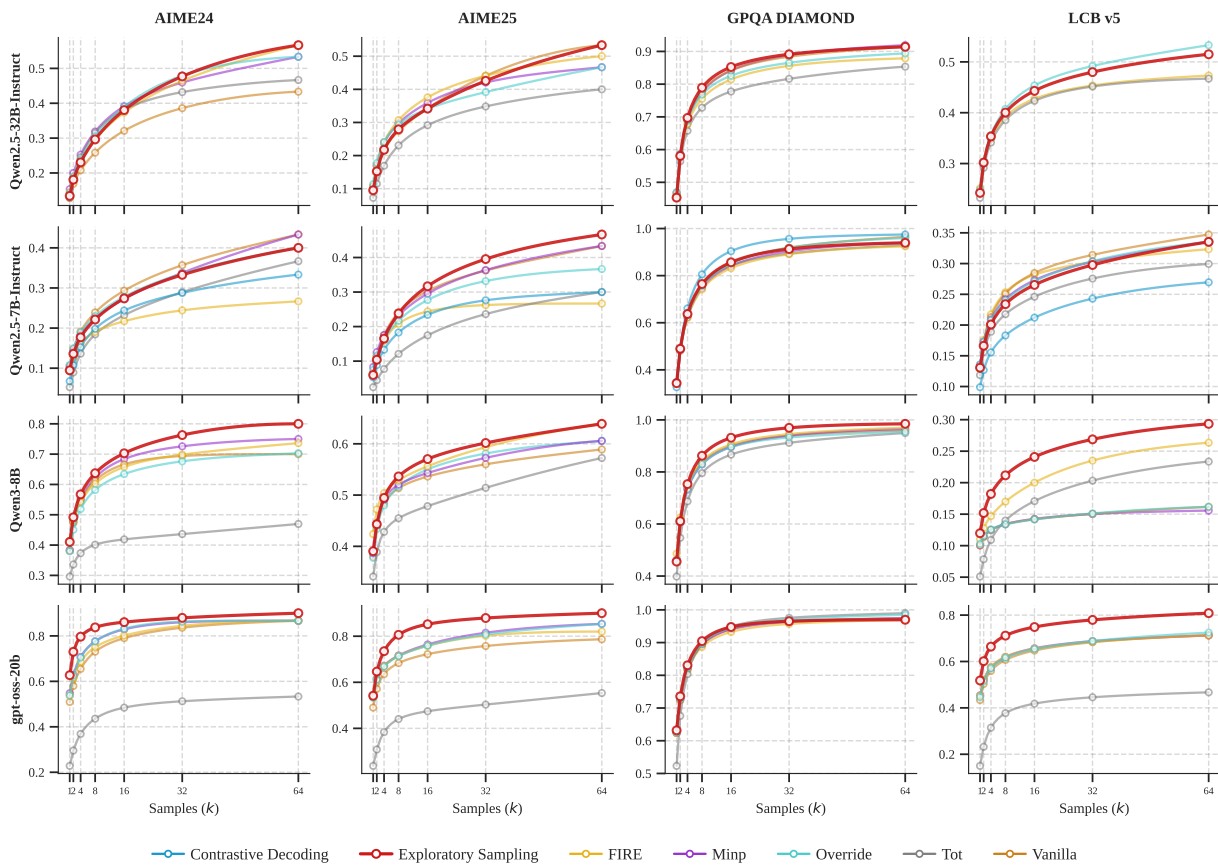

*Figure 3.* Pass@$k$ performance scaling across different models and benchmarks. `ESamp` shows superior or comparable performance to all baselines.

benchmarks. Notably, on GPT-OSS-20B, `ESamp` achieves remarkable efficiency, attaining performance comparable to the Pass@64 of baseline methods with only Pass@8. Overall, `ESamp` yields the most significant gains in math tasks (AIME24/25) compared to QA or coding. We hypothesize that the open-ended nature of mathematical reasoning allows the model to better capitalize on the semantic exploration offered by `ESamp`.

**`ESamp` Promotes Semantic Diversity.**   We analyze the trade-off between generation quality and diversity in Table 1. Using Qwen2.5-7B-Instruct, we generate 16 sequences in parallel for each prompt.

Standard decoding strategies often face a conflict between coherence and diversity. As shown in the Creative Writing results, methods like Min-P improve generation quality but restrict diversity, evidenced by lower Vendi scores and higher pairwise similarity compared to vanilla sampling. This suggests that while the outputs are coherent, they remain semantically repetitive.

`ESamp` breaks this trade-off. It achieves the highest diver-

sity and the lowest semantic similarity while simultaneously maintaining the best generation quality. This indicates that the exploration induced by `ESamp` yields candidates that are not only semantically distinct but also linguistically high-quality. In the **Math Reasoning** domain, `ESamp` similarly achieves the highest diversity scores alongside superior Pass@$k$. This confirms that our method successfully explores distinct valid reasoning paths, rather than simply generating diverse but incorrect hallucinations.

*Table 1.* Diversity and Quality Evaluation on Creative Writing and Math Reasoning (AIME25). `ESamp` consistently improves semantic diversity. Sim.: Cosine Similarity; PPL: Perplexity.

| Method | Creative Writing | | | Math (AIME) | |
|---|---|---|---|---|---|
| | Vendi ↑ | Sim. ↓ | PPL ↓ | Pass@16 ↑ | Vendi ↑ |
| Vanilla | 1.62 | 0.58 | 4.08 | 30.3% | 0.32 |
| Min-P | 1.56 | 0.62 | 3.93 | 29.5% | 0.30 |
| OverRIDE | 1.61 | 0.59 | 4.11 | 27.7% | 0.35 |
| **`ESamp` (Ours)** | **1.67** | **0.57** | **3.55** | **31.7%** | **0.46** |

*Table 2.* Sensitivity analysis of `ESamp` on AIME25 (Qwen2.5-7B-Instruct). We report Pass@$k$ (%) for $k \in \{1, 2, 4, 8, 16, 32, 64\}$. The default setting ($\beta = 0.25$, proposed fusion) is highlighted in **bold**.

| Ablation Category | Setting | Pass@1 | Pass@2 | Pass@4 | Pass@8 | Pass@16 | Pass@32 | Pass@64 |
|---|---|---|---|---|---|---|---|---|
| | $\beta = 0.1$ | 6.0% | 10.4% | 16.0% | 21.5% | 26.7% | 32.3% | 40.0% |
| Exploration Strength $\beta$ | $\beta = 0.25$ **(Default)** | 6.0% | 10.4% | 16.5% | 23.8% | 31.7% | 39.5% | 46.7% |
| | $\beta = 0.5$ | 4.8% | 8.5% | 13.4% | 18.8% | 23.8% | 28.1% | 30.0% |
| | Subtraction $(1 - \beta), \beta = 0.5$ | 0.2% | 0.4% | 0.8% | 1.7% | 3.3% | 6.7% | 13.3% |
| Fusion Formulation | Subtraction $(1 - \beta), \beta = 0.2$ | 5.8% | 10.2% | 14.8% | 19.7% | 24.6% | 28.1% | 33.3% |
| | Subtraction $(1 - \beta), \beta = 0.1$ | 6.7% | 11.2% | 15.6% | 21.3% | 29.5% | 35.1% | 40.0% |
| | **Proposed** $(1 + \beta), \beta = 0.25$ | 6.0% | 10.4% | 16.5% | 23.8% | 31.7% | 39.5% | 46.7% |

### 5.3. Decoding and Latent Distillation Dynamics

To better understand the internal behavior of `ESamp`, we analyze the generation dynamics of parallel workers and the training dynamics of the `LD`.

**Generation Trajectory Divergence.** Figure 4(a) tracks the average pairwise cosine similarity between $k$ parallel generations over decoding steps (Qwen2.5-7B on BookCorpus). Initially, all methods exhibit a rapid drop in similarity, indicating that parallel sequences quickly diverge from the common prompt. However, baseline methods soon enter a plateau phase where the decline decelerates significantly, suggesting that the diversification of candidates stagnates as the sequences lengthen. In contrast, `ESamp` maintains a continuous downward trend throughout the generation process. This sustained divergence arises because the shared Distiller acts as a centralized coordinator: once a semantic path is traversed by one sequence, it becomes "predictable" to the Distiller, naturally steering other parallel sequences toward unexplored semantic regions.

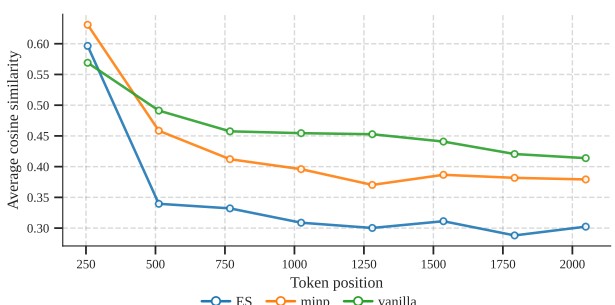

*Figure 4.* Decoding and Distillation Dynamics. `ESamp` encourages parallel generations to diverge semantically over time.

### 5.4. Efficiency Analysis

A major barrier for test-time scaling methods is the additional computational and memory overhead. We evaluate the throughput (tokens/sec) on a single RTX4090 GPU under three scenarios: (1) **Single-User** ($B = 1, K = 1$); (2) **High-Throughput** ($B = 32, K = 1$); and (3) **Test-Time Scaling** ($B = 32, K = 16$).

As shown in Table 3, the asynchronous implementation of `ESamp` introduces negligible overhead.

**Analysis of Serving Scenarios.** As shown in Table 3, the asynchronous implementation of `ESamp` incurs negligible overhead. In standard serving scenarios ($K = 1$), the overhead is less than 2%, as the Distiller's computation is fully overlapped with the LLM's execution. Crucially, in the **Test-Time Scaling scenario** ($K = 16$), the overhead rises only marginally to $\approx 4.25\%$. Unlike methods for coordinating parallel exploration with complex sequential dependencies (e.g., Tree of Thoughts), `ESamp` utilizes a shared training signal that coordinates exploration without requiring expensive cross-worker communication. This makes `ESamp` a highly practical solution for deployments.

*Table 3.* Efficiency comparison on an RTX4090 GPU (Qwen3-8B). We compare throughput across different Prompt Batches ($B$) and Samples per Request ($K$). `ESamp` maintains high efficiency even under massive parallel scaling.

| Scenario ($B \times K$) | Vanilla | `ESamp` | Overhead (%) |
|---|---|---|---|
| $B = 1, K = 1$ | 55.1 | 54.9 | 0.3% |
| $B = 32, K = 1$ | 1215.2 | 1193.2 | 1.81% |
| $B = 32, K = 16$ | 4557.7 | 4364.0 | 4.25% |

**Memory Footprint.** The memory overhead is minimal. The `LD` and its hidden state buffer consume less than 200MB of VRAM for an 8B model. With optimization, this can theoretically be reduced to around 50MB, as the distiller hidden states are low-dimensional.

### 5.5. Sensitivity Analysis

**Influence of Exploration Strength $\beta$.** We ablate the hyperparameter $\beta$, which controls the intensity of exploration. For Qwen2.5-7B-Instruct on AIME25, setting $\beta = 0.25$ yields the best balance. $\beta$ being too low causes the method to degenerate to Vanilla sampling, while values too high degrade performance by excessively penalizing high-confidence tokens.

**Logit Fusion Formulation.** We compared our fusion formula $\text{logit}_{\text{new}} = (1 + \beta)\text{logit}_{\text{ref}} - \beta\text{logit}_{\text{dist}}$ with a naive subtraction $\text{logit}_{\text{ref}} - \beta\text{logit}_{\text{dist}}$. The empirical results show

that the $(1+\beta)$ formulation preserves the relative probability mass of the base model better, preventing the model from generating grammatically incorrect sequences while still encouraging exploration.

## 6. Conclusion

We introduce Exploratory Sampling (`ESamp`) to address the limitations of standard decoding, which often yields surface-level lexical variations without genuine semantic diversity. By estimating novelty in internal representations, `ESamp` steers generation toward under-explored semantic regions. Empirical results demonstrate that `ESamp` matches or outperforms baselines, while our asynchronous pipeline ensures negligible latency overhead, establishing `ESamp` as a practical solution for efficient LLM exploration.

## Acknowledgements

The research received support from National Natural Science Foundation of China (Grant No. 62406193). We are also supported by the National Natural Science Foundation of China (No. 62506041).

The authors gratefully acknowledge further assistance provided by the Shanghai Frontiers Science Center of Human-centered Artificial Intelligence, the MoE Key Lab of Intelligent Perception and Human-Machine Collaboration, the ShanghaiTech GenAI Platform, and the HPC Platform of ShanghaiTech University.

## Impact Statement

This paper presents work whose goal is to advance the field of Machine Learning. There are many potential societal consequences of our work, none which we feel must be specifically highlighted here.

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

## A. Derivations

### A.1. Closed-Form Optimal Policy

We provide the derivation for the closed-form solution of the KL-regularized reinforcement learning objective referenced in Section 4.2.

**Problem Formulation:** We seek a policy $\pi(z|s)$ that maximizes the expected reward $r(s, z)$ while staying close to a reference policy $\pi_{\text{ref}}(z|s)$ to ensure coherence. The objective function is:

$$J(\pi) = \mathbb{E}_{z \sim \pi(\cdot|s)}[r(s, z)] - \alpha D_{\text{KL}}(\pi(\cdot|s)||\pi_{\text{ref}}(\cdot|s)) \tag{11}$$

where $\alpha > 0$ is the regularization coefficient (temperature). Expanding the KL divergence:

$$J(\pi) = \sum_z \pi(z|s)r(s, z) - \alpha \sum_z \pi(z|s) \log \frac{\pi(z|s)}{\pi_{\text{ref}}(z|s)} \tag{12}$$

We introduce a Lagrange multiplier $\lambda$ for the constraint that probabilities must sum to 1 ($\sum \pi(z|s) = 1$). The Lagrangian is:

$$\mathcal{L}(\pi, \lambda) = \sum_z \pi(z|s) \left( r(s, z) - \alpha \log \pi(z|s) + \alpha \log \pi_{\text{ref}}(z|s) \right)$$
$$- \lambda \left( \sum_z \pi(z|s) - 1 \right) \tag{13}$$

**Optimization:** Taking the derivative with respect to $\pi(z|s)$ and setting it to zero:

$$\frac{\partial \mathcal{L}}{\partial \pi(z|s)} = r(s, z) - \alpha \left( \log \pi(z|s) + 1 \right) + \alpha \log \pi_{\text{ref}}(z|s) - \lambda = 0 \tag{14}$$

Rearranging terms to solve for $\log \pi(z|s)$:

$$\alpha \log \pi(z|s) = r(s, z) + \alpha \log \pi_{\text{ref}}(z|s) - \lambda - \alpha \tag{15}$$

$$\log \pi(z|s) = \log \pi_{\text{ref}}(z|s) + \frac{1}{\alpha} r(s, z) - \frac{\lambda + \alpha}{\alpha} \tag{16}$$

Exponentiating both sides:

$$\pi^*(z|s) = \pi_{\text{ref}}(z|s) \exp \left( \frac{r(s, z)}{\alpha} \right) \cdot \exp \left( -\frac{\lambda + \alpha}{\alpha} \right) \tag{17}$$

The term $\exp(-\frac{\lambda + \alpha}{\alpha})$ acts as a normalization constant (partition function $Z(s)$) to ensure $\sum \pi^*(z|s) = 1$. Thus:

$$\pi^*(z|s) \propto \pi_{\text{ref}}(z|s) \exp \left( \frac{r(s, z)}{\alpha} \right) \tag{18}$$

$\square$

### A.2. Reward Structure and Per-Step Optimality

In Section 3, we optimize a per-step reward $r(s_t, z_t)$ in place of the sequential $Q$-function. Here we formalize the structural property under which this substitution is exact, and then discuss how the Latent Distiller approximates this ideal in practice.

**Ideal reward structure.** The following property captures the intended design of the novelty reward in our batch-parallel generation setting.

**Definition A.1.** A novelty reward $\{r_t\}_{t \geq 1}$ satisfies *vanishing redundancy* if, whenever the current prefix resides in a semantic region already explored by the batch, all continuations remain in explored regions:

$$r_t(s_t, z_t) = 0 \implies r_{t'}(s_{t'}, z_{t'}) = 0, \quad \forall t' > t, \ \forall z_{t'} \in \mathcal{V}. \tag{19}$$

**Proposition A.2.** *If the novelty reward satisfies Definition A.1, then the optimal Q-function of the sequential KL-regularized problem equals the per-step reward:* $Q^*(s_t, z_t) = r_t(s_t, z_t)$.

*Proof.* In the sequential formulation, $Q^*(s_t, z_t) = r_t(s_t, z_t) + \gamma \mathbb{E}[V^*(s_{t+1})]$.

**Case 1** ($r_t = 0$): By Definition A.1, all future rewards are zero. Hence $V^*(s_{t+1}) = 0$ and $Q^*(s_t, z_t) = 0 = r_t(s_t, z_t)$.

**Case 2** ($r_t > 0$): The current step is the first visit to a novel semantic region. After this observation, the reward for this region transitions to zero (the region has now been explored). Applying Eq. (19) from step $t + 1$ onward, all subsequent rewards within this region are zero. Therefore $V^*(s_{t+1}) = 0$ and $Q^*(s_t, z_t) = r_t(s_t, z_t)$. $\square$

Consequently, the optimal sequential policy $\pi^* \propto \pi_{\text{ref}} \exp(Q^*/\alpha)$ reduces to $\pi^* \propto \pi_{\text{ref}} \exp(r/\alpha)$, which is precisely Eq. (2).

**Latent Distiller as a practical estimator.** Definition A.1 is an idealized specification. The Latent Distiller provides a continuous approximation whose quality rests on two empirical properties.

**Assumption A.3** (Rapid Fitting). Under appropriate training configuration (e.g., aggressive learning rate, lightweight architecture), the Distiller reduces its prediction error on a newly observed representation mapping $(h_t^1 \to h_t^L)$ to near zero within a single gradient step.

**Assumption A.4** (Local Generalization). After fitting a mapping $(h_t^1 \to h_t^L)$, the Distiller maintains low prediction error for neighboring mappings $(h_{t'}^1 \to h_{t'}^L)$ with $\|h_{t'}^1 - h_t^1\| < \epsilon$.

Together, these properties ensure that the Distiller's prediction error approximates the vanishing redundancy property: once a semantic region is explored, the Distiller assigns near-zero novelty to that region and its neighborhood, mirroring the behavior specified in Definition A.1.

**Adaptivity beyond the ideal.** Notably, the Distiller does not strictly satisfy Definition A.1, and this is beneficial. The ideal specification is rigid: once a region is marked as explored, it remains so permanently. The Distiller, by contrast, evaluates novelty at each step via its current prediction error. If a trajectory whose prefix resides in a well-explored region later encounters a genuinely novel representation mapping (one far from any previously observed mapping), the Distiller will detect this and assign a positive reward. This makes the Distiller a soft relaxation of the ideal: it suppresses redundancy where it persists, while remaining responsive to genuine novelty whenever it emerges.

**Empirical support.** Figure 4 corroborates both assumptions: the monotonically decreasing pairwise cosine similarity under `ESamp` confirms that semantic divergence induced at early steps is sustained throughout generation (consistent with local generalization), while the Distiller's rapid adaptation to newly observed patterns supports the rapid fitting assumption.

# B. Engineering the Parallel Pipeline

To achieve the virtually zero-overhead behavior described in Section 4.4, we implement a custom runtime environment bypassing the standard Python Global Interpreter Lock (GIL) limitations during the forward pass. The implementation relies on three key engineering pillars:

## B.1. Asynchronous CUDA Streams

We establish a dual-stream architecture. The `Main Stream` handles the standard vLLM execution graph. A dedicated high-priority `Distiller Stream` handles the MLP computations. Synchronization is managed strictly via CUDA Events (`cudaEventRecord` and `cudaEventWait`). Unlike CPU barriers (e.g., `torch.cuda.synchronize()`), CUDA Events operate directly on the GPU command processor. This allows the CPU to dispatch the entire instruction chain for both streams without halting, effectively hiding the kernel launch overheads.

## B.2. Pre-allocated "In-Graph" Memory

Dynamic memory allocation (`malloc`) is a significant source of latency. We implement a static GPU ring buffer to manage hidden states.

- Write Operation: We inject a custom kernel at the output of the LLM's first layer to write the hidden states directly to the buffer.

- Read Operation: The Distiller reads from this buffer asynchronously.

Crucially, no data is copied to the CPU RAM at any stage of the pipeline. All operations, including the distillation loss calculation and weight updates, occur in high-bandwidth GPU memory (HBM), avoiding the PCIe bottleneck.

### B.3. Mixed-Batch Guardrails

In production environments using continuous batching (e.g., vLLM), a single batch may contain both prefill (prompt processing) and decode (token generation) requests. Since our Distiller is optimized for the generation phase, we implement a lightweight metadata check. The pipeline activates only for decode-phase tokens. For mixed batches or prefill-heavy steps, the Distiller is dynamically bypassed. Given that autoregressive generation typically spans hundreds of steps compared to a single prefill step, this selective bypassing has negligible impact on model convergence.

### B.4. Latency Analysis

The "slack" available for Distiller inference is determined by the execution time of layers to . For a standard Llama-3-8B model on an A100 GPU, this interval is approximately 15-20ms. The Distiller (a 2-layer MLP) requires less than 0.5ms. This massive margin guarantees that the Distiller's logits are always available before the logit fusion is launched.

## C. Additional Analyses

In this session, we try to answer several questions about the sensitivity of `ESamp`, the validity of the latent novelty signal, statistical stability, baseline fairness, composability with other decoding or aggregation methods, and the engineering overhead of the practical implementation. We summarize the additional experiments and clarifications here. These results complement the main paper without changing the core method or the conclusions.

### C.1. Hyperparameter Sensitivity Across Model Scales

`ESamp` introduces one primary exploration-strength hyperparameter, $\beta$. The hidden-layer choice is not treated as a tunable hyperparameter in our implementation: the Distiller is architecturally fixed to map the first-layer hidden state to the final-layer hidden state. This design is dictated by the asynchronous pipeline in Section 4.4. Using the earliest available hidden state maximizes the overlap window between the Distiller computation and the remaining LLM forward pass, which is crucial for maintaining low runtime overhead.

To examine whether $\beta$ requires model-specific tuning, we evaluate three values of $\beta$ across the Qwen3 model family on AIME24. The results are shown in Table 4. A single setting, $\beta = 0.25$, performs consistently well across model scales.

*Table 4.* Sensitivity of `ESamp` to $\beta$ across Qwen3 model scales on AIME24.

| Model | Metric | $\beta = 0.1$ | $\beta = 0.25$ | $\beta = 0.4$ |
|---|---|---|---|---|
| Qwen3-4B | Pass@16 / Pass@64 | 70.7 / 83.3 | 68.9 / 80.0 | 68.7 / 76.7 |
| Qwen3-8B | Pass@16 / Pass@64 | 68.7 / 76.7 | 70.0 / 80.0 | 67.3 / 76.7 |
| Qwen3-14B | Pass@16 / Pass@64 | 69.6 / 76.7 | 72.2 / 83.3 | 70.5 / 76.7 |

The results suggest that `ESamp` is not highly sensitive to small changes in $\beta$. In particular, $\beta = 0.25$ provides a robust default without per-model tuning. We use this value in the main experiments unless otherwise specified.

### C.2. Pass@1 Accuracy and the Exploration–Grounding Trade-off

A natural concern is that encouraging exploration may reduce per-sample accuracy, especially if the novelty signal were to push generation toward less grounded continuations. `ESamp` avoids this failure mode by preserving the base LLM's full forward computation. The final-layer representation and logits are computed exactly as in standard decoding. The Distiller does not replace the LLM hidden state and does not directly generate tokens from shallow-layer representations; it only

predicts the final-layer hidden state from the shallow-layer hidden state and uses the prediction discrepancy to reweight the LLM's own candidate-token logits.

To make this point empirically explicit, we report Pass@1 results across the main reasoning and coding benchmarks. Pass@1 measures average single-sample accuracy and is therefore a direct indicator of whether `ESamp` degrades per-sample correctness.

*Table 5.* Pass@1 on AIME25.

| Model | Vanilla | Min-p | FIRE | OverRIDE | `ESamp` |
|---|---|---|---|---|---|
| Qwen2.5-7B | 6.6 | 8.3 | 6.5 | 7.4 | 6.0 |
| Qwen3-8B | 38.0 | 38.6 | 42.4 | 37.8 | 39.1 |
| GPT-OSS-20B | 49.0 | 53.8 | 53.5 | 53.3 | 54.2 |

*Table 6.* Pass@1 on AIME24.

| Model | Vanilla | Min-p | FIRE | OverRIDE | `ESamp` |
|---|---|---|---|---|---|
| Qwen2.5-7B | 10.7 | 10.3 | 9.2 | 10.8 | 9.5 |
| Qwen3-8B | 38.1 | 38.4 | 40.4 | 38.0 | 41.0 |
| GPT-OSS-20B | 57.2 | 58.7 | 57.8 | 58.8 | 62.7 |

*Table 7.* Pass@1 on LiveCodeBench v5 with context length 4096.

| Model | Vanilla | Min-p | FIRE | OverRIDE | `ESamp` |
|---|---|---|---|---|---|
| Qwen2.5-7B | 13.0 | 13.5 | 12.9 | 13.6 | 13.1 |
| Qwen3-8B | 10.0 | 10.0 | 11.0 | 10.3 | 12.0 |
| GPT-OSS-20B | 43.4 | 45.4 | 45.2 | 44.6 | 51.8 |

`ESamp` matches or exceeds Vanilla in the majority of settings. We do observe small Pass@1 decreases in a few cases, such as Qwen2.5-7B on AIME25. This is consistent with the fact that `ESamp` primarily targets candidate-set coverage rather than optimizing the single most likely sample. However, the results do not show a systematic decline in per-sample accuracy. In several settings, `ESamp` substantially improves Pass@1, for example GPT-OSS-20B on AIME24 and LiveCodeBench v5.

### C.3. Does Latent Prediction Error Encode Structured Novelty?

The central mechanism of `ESamp` is that Distiller prediction error in representation space provides a useful novelty signal. To test whether the improvement is caused by structured information in the error vector rather than arbitrary perturbation, we replace the true Distiller error vector with a Gaussian vector of matched magnitude while keeping the rest of the decoding procedure unchanged. Table 10 reports the results on Qwen3-8B.

The random-noise variant collapses to approximately Vanilla-level performance, whereas the true `ESamp` error vector produces substantial gains. This supports the interpretation that `ESamp` is not merely injecting noise into decoding. Rather, the error direction contains structured information about representation-space patterns that the online Distiller has not yet fit.

### C.4. Latent-Space Versus Vocabulary-Space Novelty

To isolate the benefit of estimating novelty in representation space, we construct a vocabulary-space Distiller baseline. This variant uses the same MLP structure, but projects through the frozen LM head and is trained online with a KL objective in vocabulary space. The logit fusion follows the same form as `ESamp`, so the main difference is the space in which novelty is estimated.

The vocabulary-space variant is unstable and substantially underperforms latent-space `ESamp`. We attribute this to the difficulty of online learning over the high-dimensional discrete vocabulary distribution, where KL gradients can be noisy and sensitive to the tail of the distribution. In contrast, `ESamp` operates in a compact continuous representation space, making the online Distiller both more stable and more useful for exploration.

*Table 8.* Pass@1 on LiveCodeBench v5 with context length 16384.

| Model | Vanilla | Min-p | FIRE | OverRIDE | ESamp |
|---|---|---|---|---|---|
| Qwen2.5-7B | 13.0 | 13.5 | 12.9 | 13.6 | 13.1 |
| Qwen3-8B | 19.2 | 19.3 | 18.3 | 20.7 | 22.6 |
| GPT-OSS-20B | 52.2 | 52.5 | 51.2 | 57.6 | 61.5 |

*Table 9.* Pass@1 on GPQA-Diamond.

| Model | Vanilla | Min-p | FIRE | OverRIDE | ESamp |
|---|---|---|---|---|---|
| Qwen2.5-7B | 34.1 | 34.1 | 34.6 | 34.8 | 34.3 |
| Qwen3-8B | 45.9 | 46.3 | 48.6 | 46.0 | 46.6 |
| GPT-OSS-20B | 62.4 | 62.4 | 62.3 | 62.5 | 63.2 |

### C.5. Multi-Seed Stability

Because `ESamp` is a stochastic decoding method, we additionally report three-seed results on Qwen3-8B using seeds 41, 42, and 43. Table 12 reports the mean and standard deviation.

The results are stable across seeds. On AIME25, `ESamp` trades lower Pass@8 for stronger Pass@32 and Pass@64, which reflects its intended use case: improving coverage when multiple candidates are sampled for test-time scaling.

### C.6. Baseline Hyperparameter Tuning

To ensure comparable tuning effort, we use three-point grids for the main decoding baselines. Table 13 reports the AIME24 Pass@64 results on Qwen3-8B.

Even the weakest `ESamp` setting matches or exceeds the best-tuned baseline settings in this comparison, while the default $\beta = 0.25$ achieves the best result.

### C.7. Surface Entropy Versus Latent Prediction Error

To compare `ESamp` against a surface-level diversity signal, we add an entropy-adaptive decoding baseline that adjusts the candidate set based on each token candidate's contribution to the normalized entropy of the token distribution. This is representative of methods that operate directly on vocabulary-space uncertainty.

`ESamp` achieves higher diversity than the entropy-adaptive baseline. This suggests that representation-space novelty captures semantic variation that is not fully captured by token-level entropy alone.

### C.8. Composability with FIRE and Self-Consistency

`ESamp` operates during decoding, while many test-time scaling methods operate either by changing the sampling schedule or by aggregating completed candidate answers. This makes `ESamp` naturally composable with other methods.

First, we combine `ESamp` with FIRE on Qwen3-8B / AIME24. FIRE modifies the temperature schedule, while `ESamp` modifies candidate-token preferences using representation-space novelty.

The combination improves Pass@64 beyond either method alone, indicating that `ESamp` is not tied to a particular temperature schedule and can complement other decoding-time interventions.

Second, we evaluate compatibility with Self-Consistency (SC), which aggregates completed answers by majority voting. `ESamp` promotes candidate diversity, whereas SC rewards convergence toward the most frequent answer, so the two objectives are not perfectly aligned. Nevertheless, `ESamp` remains compatible with SC and provides slight improvements at larger budgets.

These results suggest that `ESamp` is especially well suited to selection-based test-time scaling, such as Pass@k evaluation or reward-model reranking, where diversity directly improves solution coverage. Its interaction with majority-vote aggregation

*Table 10.* Noise ablation on Qwen3-8B. Magnitude-matched random noise does not reproduce the gains of `ESamp`, suggesting that the direction of the Distiller error vector carries structured information.

| Method | AIME25 Pass@16 | AIME25 Pass@64 | AIME24 Pass@16 | AIME24 Pass@64 |
|---|---|---|---|---|
| Vanilla | 53.6 | 58.9 | 66.6 | 70.0 |
| `ESamp` + Random Noise | 52.4 | 60.0 | 66.6 | 70.0 |
| `ESamp` | 57.0 | 63.9 | 70.0 | 80.0 |

*Table 11.* Latent-space `ESamp` compared with a vocabulary-space Distiller on Qwen3-8B.

| Method | AIME25 Pass@16 | AIME25 Pass@64 | AIME24 Pass@16 | AIME24 Pass@64 |
|---|---|---|---|---|
| Vanilla | 53.6 | 58.9 | 66.6 | 70.0 |
| Vocab-Space Distiller | 37.4 | 43.3 | 47.6 | 53.3 |
| OverRIDE | 55.0 | 60.6 | 63.4 | 70.0 |
| `ESamp` (Latent) | 57.0 | 63.9 | 70.0 | 80.0 |

is more conservative but still non-negative at larger budgets.

### C.9. Distiller Architecture Robustness

We ablate the Distiller architecture on Qwen3-8B / AIME25. The default Distiller is a two-layer Gated SwiGLU MLP. We compare it with deeper and simpler alternatives.

The Pass@k results are robust across architectures. We therefore choose the two-layer Gated SwiGLU Distiller as the default because it provides the same high-budget coverage with lower computational cost.

### C.10. Shared Versus Per-Prompt Distillers

We also compare shared and per-prompt Distiller variants. In the shared setting, a single Distiller is updated from all prompts in a batch. In the per-prompt setting, each prompt maintains its own Distiller. Table 18 shows that the preferred setting can depend on task structure.

Per-prompt Distillers perform better on AIME, where different problems can have heterogeneous reasoning structures and a shared Distiller may introduce cross-prompt interference. On LiveCodeBench, the shared Distiller slightly improves Pass@16, likely because the larger effective batch provides a stronger online learning signal. We view adaptive sharing strategies as a promising direction for future work.

### C.11. LLM-as-Judge Evaluation for Creative Writing

Automatic diversity metrics may not fully capture whether generations differ in meaningful creative content. We therefore add a single-blind LLM-as-judge evaluation on 2,000 BookCorpus prompts with Gemini 3 Flash. The judge compares 16 parallel generations per prompt and reports average diversity and quality ranks, where lower is better. The results are in figure 19.

`ESamp` obtains the best diversity rank while maintaining quality close to Vanilla. This supports the conclusion that `ESamp` improves meaningful variation rather than merely increasing surface-level randomness.

## D. The tLLM Framework and Practical `ESamp` Optimizations

The implementation used in the main paper was an internal research prototype designed to validate the algorithmic idea of `ESamp`. After the paper experiments, we further engineered an open-source implementation, `ESamp`, in the tLLM framework:



https://github.com/LinesHogan/tLLM.



The open-source version achieves higher measured throughput than the internal implementation reported in the main paper, while deliberately preserving a general interface for future runtime-adaptation algorithms.

*Table 12.* Three-seed results on Qwen3-8B.

| Benchmark | Method | Pass@8 | Pass@16 | Pass@32 | Pass@64 |
|---|---|---|---|---|---|
| AIME24 | Vanilla | $61.2 \pm 0.1$ | $66.6 \pm 0.8$ | $69.4 \pm 1.5$ | $70.0 \pm 3.3$ |
| AIME24 | ESamp | $61.8 \pm 0.2$ | $68.5 \pm 0.8$ | $74.6 \pm 1.5$ | $80.0 \pm 0.0$ |
| AIME25 | Vanilla | $51.4 \pm 1.0$ | $53.6 \pm 0.6$ | $56.0 \pm 1.0$ | $58.9 \pm 0.9$ |
| AIME25 | ESamp | $46.2 \pm 0.9$ | $53.6 \pm 1.0$ | $61.3 \pm 0.4$ | $67.8 \pm 1.9$ |

*Table 13.* Baseline hyperparameter grids on AIME24 with Qwen3-8B.

| Method | Setting 1 | Setting 2 | Setting 3 |
|---|---|---|---|
| Min-p, $p_{\text{base}} \in \{0.03, 0.1, 0.3\}$ | 73.3 | 76.7 | 76.7 |
| OverRIDE, $\lambda \in \{0.6, 0.8, 1.0\}$ | 73.3 | 76.7 | 76.7 |
| FIRE, $T \in \{10, 30, 50\}$ | 73.3 | 73.3 | 73.3 |
| ESamp, $\beta \in \{0.1, 0.25, 0.4\}$ | 76.7 | 80.0 | 76.7 |

## D.1. tLLM as a Runtime Layer for Test-Time Intervention

tLLM is a runtime layer built on top of the vLLM v1 inference engine. Its goal is to support test-time algorithms that need to read model-internal states, run asynchronous side computation, and optionally guide sampling during generation, without requiring researchers to maintain a private fork of vLLM.

The framework exposes a producer–consumer abstraction. Producers capture runtime tensors and metadata from the vLLM execution path, such as hidden states and request-row localization information. Consumers declare the data they need and receive runtime bundles during generation. This design makes it possible to implement algorithms such as ESamp as external consumers rather than by directly rewriting the vLLM worker, model runner, or sampler internals.

For ESamp, the consumer captures shallow and deep hidden states, trains the online Distiller during generation, and optionally modifies candidate-token logits after the base sampler filtering stage. This preserves the core ESamp algorithm while making the implementation easier to inspect, benchmark, and extend.

## D.2. Engineering Optimizations in ESamp

The open-source ESamp implementation includes several engineering optimizations beyond the internal research prototype used for the main-paper throughput measurement.

**Runtime hooks without maintaining a vLLM fork.** tLLM installs hooks around key vLLM lifecycle points, including model loading, input preparation, model execution, logit computation, and sampling. This allows ESamp to capture hidden states and provide sampler guidance through a stable runtime interface rather than through invasive modifications of vLLM internals.

**Producer–consumer data delivery.** Captured tensors are localized to the active request rows and delivered to the ESamp consumer through runtime bundles. This avoids ad hoc data plumbing and makes the hidden-state path explicit. It also enables functional counters, such as Distiller loss counts and sampler-guidance counters, which may help third-party developers to ensure that throughput benchmarks are not accidentally measuring a disabled or no-op algorithm.

**Post-filter candidate intervention.** Instead of projecting the Distiller prediction across the full vocabulary on the sampler hot path, ESamp can apply the intervention only to the candidate tokens retained after the LLM sampler's filtering stage. The implemented intervention follows the same form as the paper. Restricting the intervention to the filtered candidate set substantially reduces overhead while preserving the intended ESamp behavior.

**CUDA graph support for Distiller updates.** The optimized benchmark path supports CUDA graph capture for the Distiller update. This reduces repeated kernel-launch overhead in the online training loop and makes the Distiller update path more predictable during autoregressive generation.

*Table 14.* Comparison with an entropy-adaptive decoding baseline on Qwen2.5-7B-Instruct. Higher Vendi and lower similarity indicate greater semantic diversity.

| Method | Vendi ↑ | Similarity ↓ |
|---|---|---|
| Vanilla | 1.6403 | 0.5845 |
| Entropy-Adaptive | 1.6455 | 0.5830 |
| ESamp | 1.6698 | 0.5713 |

*Table 15.* Composing `ESamp` with FIRE on Qwen3-8B / AIME24.

| Method | Pass@1 | Pass@4 | Pass@16 | Pass@64 |
|---|---|---|---|---|
| Vanilla | 38.1 | 54.6 | 66.7 | 70.0 |
| FIRE | 41.4 | 54.2 | 65.8 | 73.6 |
| ESamp | 41.0 | 56.7 | 70.2 | 80.0 |
| FIRE + ESamp | 38.4 | 54.4 | 69.0 | 83.3 |

**Triton grouped prediction backend.** For CUDA/Qwen throughput experiments, `ESamp` provides an optional Triton grouped backend for the no-gradient Distiller prediction and sampling path. Training and autograd can remain in PyTorch, while the latency-sensitive prediction path benefits from a more specialized GPU implementation.

**Compatibility with modern vLLM inference optimizations.** The open-source implement is aligned with a modern vLLM V1 baseline using CUDA Graph execution, FlashInfer sampling, bfloat16 weights, prefix-cache control for fair measurement, and the same sampling workload. `ESamp` is designed to preserve these inference-engine optimizations rather than replacing them with a slower research-only runtime.

### D.3. Open-Source Throughput

Table 20 reports the representative open-source throughput benchmark from the optimized `ESamp` path. The benchmark uses Qwen2.5-7B, batch size 8, $n = 16$, and an active min-p sampling path on an RTX 4090 GPU. Throughput is measured against an optimized vLLM baseline under the same workload.

This corresponds to approximately 98.8% of the optimized vLLM baseline throughput in the aligned open-source benchmark. Equivalently, the measured throughput reduction is about 1.2%. This number is higher than the conservative internal throughput result reported in the main paper.

We emphasize that the open-source implementation is not a fully task-specialized kernel-only implementation. It intentionally preserves a general producer–consumer interface, runtime validation counters, configurable Distiller paths, and extension points for future test-time learning algorithms. If one removes these generality and extensibility constraints, further throughput improvements should be possible. Therefore, the throughput in Table 20 should be interpreted as a practical open-source framework result rather than an absolute upper bound on `ESamp` efficiency.

### D.4. Scope of the Open-Source Implementation

The main paper focuses on the algorithmic contribution of `ESamp`: using online latent distillation to encourage representation-space exploration during decoding. tLLM and `ESamp` provide the accompanying systems path for reproducing and extending this idea in a high-throughput inference engine. Beyond `ESamp`, the same framework can support other test-time intervention methods, including activation analysis, hidden-state export or editing, online auxiliary-model training, and candidate-level sampler guidance.

This separation is intentional. `ESamp` defines the decoding objective and novelty signal, while tLLM provides a practical runtime substrate for implementing `ESamp`-like algorithms under realistic serving constraints.

*Table 16.* Composing `ESamp` with Self-Consistency on Qwen3-8B / AIME24.

| Method | Maj@8 | Maj@16 | Maj@32 |
|---|---|---|---|
| Vanilla + SC | 50.3 | 52.6 | 53.7 |
| `ESamp` + SC | 50.0 | 52.8 | 54.5 |

*Table 17.* Distiller architecture ablation on Qwen3-8B / AIME25.

| Architecture | Pass@1 | Pass@16 | Pass@64 |
|---|---|---|---|
| 2-layer Gated SwiGLU (default) | 41.0 | 66.5 | 80.0 |
| 4-layer Gated SwiGLU | 38.9 | 66.7 | 80.0 |
| 4-layer Plain MLP | 38.9 | 66.7 | 80.0 |

# E. Experiment Details

## E.1. Benchmarks and Metrics

**Benchmarks**    In our main experiments, we evaluated our method on **AIME 2024**, **AIME 2025**, **LiveCodeBench v5**, and **GPQA-Diamond**. All benchmarks were evaluated using the `lighteval` framework (Habib et al., 2023). We utilized the official data and evaluation code provided by `lighteval`. The benchmarks are detailed as follows:

- **AIME 2024 / 2025**: The American Invitational Mathematics Examination (AIME (Zhang & Math-AI, 2024; 2025)) datasets serve as a standard for evaluating advanced mathematical reasoning. These problems are designed for high-performing high school students and require multi-step logical deduction to reach a final integer answer between 0 and 999. We use both the 2024 and 2025 iterations to assess the model's performance on recent, complex mathematical tasks.

- **LiveCodeBench v5**: This is a holistic benchmark for code generation (Jain et al., 2024) that evaluates models on competitive programming problems collected from platforms like LeetCode, AtCoder, and Codeforces. A key feature of LiveCodeBench is its time-sensitive nature; it specifically targets problems released after the training cutoff of most large language models to minimize data contamination. We employ the v5 release to test the model's ability to generate correct and efficient Python code for novel problem specifications.

- **GPQA-Diamond**: The Graduate-Level Google-Proof Q&A (GPQA) benchmark (Rein et al., 2024) consists of challenging multiple-choice questions covering biology, physics, and chemistry. We utilize the "Diamond" subset, which represents the highest quality tier where questions have been double-verified by domain experts. This benchmark is designed to be resistant to simple information retrieval, requiring deep scientific reasoning and domain knowledge to solve.

To evaluate diversity in creative writing, we incorporated the **BookCorpus** dataset (Zhu et al., 2015). We used the cleaned version provided by `incredible45/Gutenberg-BookCorpus-Cleaned-Data-English` on HuggingFace, which contains approximately 51,442 classic literary works. Since this dataset lacks explicit source text delimiters (e.g., headers or metadata like `robots.txt`), using the first few tokens as a prompt often results in non-narrative generation that fails to meet creative writing evaluation standards. To address this, we split each text in the middle and used the last 32 words of the first half as the prompt to generate the continuation without using a chat template. Given the sheer size of the dataset, evaluating the full set was computationally prohibitive. Therefore, we sampled 2,000 stories using a fixed random seed (`seed=42`) for generation and evaluation.

**Evaluation Metrics**    We employed the following metrics to assess performance and diversity:

- **Pass@k**: The probability that at least one correct solution exists among $k$ generated samples.

- **Embedding Similarity**: The average pairwise cosine similarity of the embeddings of the generated responses, measuring semantic redundancy.

*Table 18.* Shared versus per-prompt Distillers.

| Benchmark | Shared | Per-Prompt |
|---|---|---|
| AIME24 Pass@16 / Pass@64 | 64.7 / 76.6 | 70.0 / 80.0 |
| AIME25 Pass@16 / Pass@64 | 55.3 / 63.3 | 57.0 / 63.9 |
| LCB v5 Pass@16 | 25.8 | 24.1 |

*Table 19.* Single-blind LLM-as-judge evaluation on BookCorpus prompts. Lower rank is better.

| Method | Diversity Rank ↓ | Quality Rank ↓ |
|---|---|---|
| Vanilla | 1.97 | 1.83 |
| OverRIDE | 2.40 | 2.20 |
| ESamp | 1.63 | 1.97 |

- **Vendi Score**: A spectral metric for diversity, calculated using the official implementation from `https://github.com/vertaix/Vendi-Score`.

- **Perplexity (PPL)**: Used as a proxy for linguistic fluency and coherence.

### E.2. Implementation Details

The code will be released upon the acceptance of this paper.

**Exploratory Sampling Configuration**  The Latent Distiller is implemented as a two-layer Multi-Layer Perceptron (MLP) with residual connections. Each layer consists of a Gated SwiGLU block, where the input and output dimensions align with the host LLM's hidden size, while the intermediate hidden dimension is set to 384. Regarding the hyperparameters, for inference, the exploration strength was set to $\beta = 0.25$. For training, we used the Adam optimizer with a learning rate of $4 \times 10^{-4}$, $\epsilon = 1 \times 10^{-4}$, and a gradient clipping norm of $0.5$. These relatively aggressive optimization settings were chosen because the distiller operates in an online learning scenario, where the distribution of training data and inference data may shift rapidly.

An interesting empirical observation is related to the distiller's scope. Although theoretical formulation suggests that each prompt should maintain an independent distiller to capture its specific trajectories, we found in practice—specifically during AIME development—that sharing a single distiller across the batch did not yield significant performance differences. We hypothesize this is because the mathematical problems in AIME are highly distinct, preventing the "experience" from one problem's distiller update from interfering with another. Future work could exploit this property to isolate the training influence of each prompt while optimizing memory usage, potentially reducing the overhead of maintaining multiple distiller states.

**vLLM System Details**  All experiments except the throughput experiment were conducted on NVIDIA A100 GPUs. To minimize latency and maximize throughput, we implemented several engineering optimizations:

- **Latent Space Logit Mixing**: We moved the "logit mixing" operation to a stage before the vLLM sampler's filtering (e.g., top-p, min-p) if filter not set. This allows us to exploit the distributive property of matrix multiplication. Specifically, instead of projecting both the reference hidden state $h_{ref}$ and the distilled hidden state $h_{dist}$ to the vocabulary space separately (which requires two large matrix multiplications with the LM Head), we mix them in the latent space: $h_{mix} = (1 + \beta)h_{ref} - \beta h_{dist}$. We then perform a single projection $h_{mix}W_{head}$. This optimization reduces the computational cost of the LM Head projection.

- **Batched Distiller Operations**: In scenarios with static throughput (fixed computation graph), we batch the inference and update steps of multiple distillers. Since the distiller model is extremely lightweight (consuming less than 10MB of VRAM), the overhead of launching individual CUDA kernels is non-negligible compared to the computation itself. Batching these operations effectively amortizes the kernel launch overhead.

*Table 20.* Representative open-source `ESamp` throughput in tLLM. The optimized `ESamp` implementation preserves most of the vLLM baseline throughput while running the `ESamp` runtime-adaptation path.

| Model / Workload | Optimized vLLM Baseline | `ESamp` (Triton Kernel) | Ratio |
|---|---|---|---|
| Qwen2.5-7B, batch=8, $n = 16$, min-p active path | 5370.616 tok/s | 5304.855 tok/s | 0.9878 |

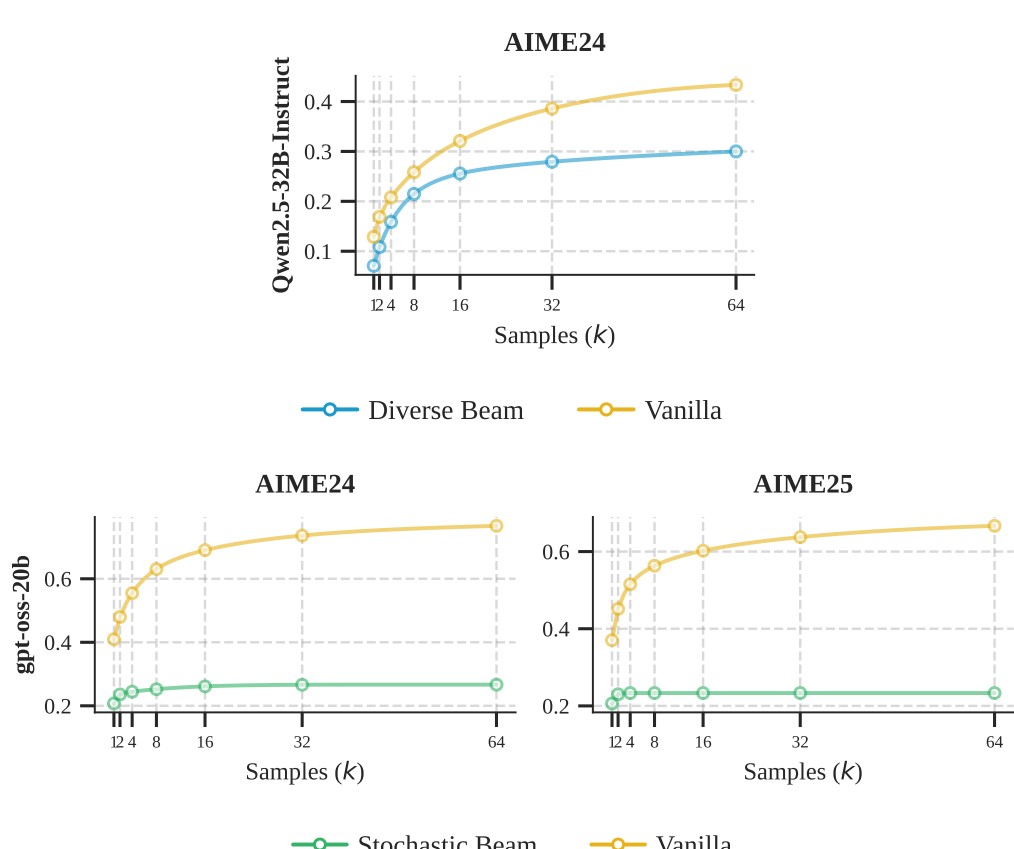

*Figure 5.* Experiment results in beam search of Pass@k

### E.3. Beam Search Results

Beam search is fundamentally designed to find the sequence with the maximum joint probability. However, this objective is often antithetical to the goal of exploration required for reasoning tasks. In our preliminary experiments, both Diverse Beam Search and Stochastic Beam Search exhibited poor performance on the Pass@k metric compared to sampling-based methods. Consequently, we have omitted detailed beam search results from the main paper to avoid clutter and potential confusion regarding the efficacy of exploration strategies.

### E.4. OverRIDE and Contrastive Decoding

For Contrastive Decoding, we employ Qwen2.5-0.5B-Instruct as the amateur model for the Qwen2.5 series, and Qwen3-0.6B for the Qwen3 series. We omit experiments on GPT-OSS-20B as no corresponding small-scale model shares its vocabulary. Regarding OverRIDE, we adopt the hyperparameter configurations specified in its OpenReview Supplementary Material, setting $\lambda = 0.8$, the number of iterations to 10, the adapter rank to 16, and the learning rate to $10^{-3}$.

## E.5. Prompts

We utilized the official prompts provided by the `lighteval` framework for each benchmark. The specific templates are listed below:

### AIME 2024 / AIME 2025

```
Solve the following math problem efficiently and clearly.  The last line of
your response should be of the following format: 'Therefore, the final
answer is: $\boxed{ANSWER}$. I hope it is correct' (without quotes) where
ANSWER is just the final number or expression that solves the problem.
Think step by step before answering.

{Question}
```

### LiveCodeBench (Code Generation)

```
You will be given a question (problem specification) and will generate a
correct Python program that matches the specification and passes all tests.

Question: {question_content}
```

If `starter_code` is provided:

```
You will use the following starter code to write the solution to the
problem and enclose your code within delimiters.
```python
{starter_code}
```
```

If no `starter_code` is provided:

```
Read the inputs from stdin solve the problem and write the answer to stdout
(do not directly test on the sample inputs). Enclose your code within
delimiters as follows. Ensure that when the python program runs, it reads
the inputs, runs the algorithm and writes output to STDOUT.
```python
# YOUR CODE HERE
```
```

### GPQA

```
Answer the following multiple choice question. The last line of your response
should be of the following format: 'Answer: $LETTER' (without quotes) where
LETTER is one of ABCD. Think step by step before answering.

{Question}

A) {A}
B) {B}
C) {C}
D) {D}
```

### Tree of Thoughts (self-review system prompt)

```
You are a Tree-of-Thought {task type, such as mathematics} researcher.
```

```
Explore multiple distinct reasoning branches in parallel,
keep them concise, and prune clearly wrong branches.
Summarize the surviving ideas into a final, step-by-step
solution before producing the final numeric answer. Always end your
response with "Final Answer: <number>" where the number is a single
integer between 0 and 999.
```

### E.6. Case Study

Due to the very large volume of parallel generation, even if a single mathematical solution only contains 1000 tokens, the total number of tokens across 16 answers exceeds the total number of words in this document. Therefore, we provide an evaluation of the two sets of answers generated by vanilla generation and `ESamp` generation using Gemini 3 Flash Preview. This is a single-blind test; Gemini 3 Flash Preview does not know which generation method it is evaluating. File 1 is generated by `ESamp` while file 2 is by vanilla generation with Qwen2.5-7B-Instruct.

**Evaluation Output by Gemini 3 Flash Preview:**

## 1. Problem Statement
The problem involves a 9-kilometer walk with a coffee shop stop of $t$ minutes. We are given two scenarios to find the relationship between walking speed $s$ (km/h) and total time:

- Scenario 1: Speed $s \implies$ Total time = 4 hours.
- Scenario 2: Speed $s + 2 \implies$ Total time = 2 hours and 24 minutes (2.4 hours).

The goal is to determine the total time required when the walking speed is $s + \frac{1}{2}$ km/h.

## 2. Analysis of File 1
File 1 demonstrates significant variety in mathematical modeling and execution strategies across its sequences.

*Methodological Approaches*

- **Algebraic Elimination (Standard):** Most sequences solve by subtracting the two distance/time equations to eliminate the variable $t$ and solve for $s$ first.
- **Direct Quadratic for $t$:** Sequences 3 and 14 attempt to solve for the coffee shop time $t$ as the primary variable. This leads to more complex polynomial expansions ($t^2 - 384t + 8640 = 0$), representing a different logical priority.
- **Heuristic / Trial and Error:** Sequence 11 stands out by using a "guess and check" strategy. It assumes plausible values for $t$ (e.g., 60, 48, 30, 24 minutes) and verifies them against the equations. This mimics human intuition rather than rote computation.
- **Unit Management:** There is a mix of working in pure minutes, pure hours, or hybrid fractional forms. Some sequences use $x = t/60$ as a placeholder, while others carry $t$ through large-scale multiplication (e.g., $540 = s(240 - t)$).

*Observations*
File 1 displays "computational personality." One sequence even shows a self-correction process after encountering a negative discriminant, which illustrates a non-linear problem-solving path.

## 3. Analysis of File 2
File 2 is highly consistent and follows a standardized "textbook" efficiency.

*Methodological Approaches*

- **Convergent Subtraction Path:** Almost every sequence (Seq 0 through Seq 15) adopts the exact same strategy: convert 2h 24m to 2.4h, set up $9/s + t_{\text{hours}} = 4$ and $9/(s + 2) + t_{\text{hours}} = 2.4$, and subtract to find $s$.
- **Uniform Quadratic Formulation:** All sequences converge on the quadratic $s^2 + 2s - 11.25 = 0$.
- **Linear Logic:** The sequences rarely deviate from the order of: 1) Solve $s$, 2) Solve $t$, 3) Solve final time.

*Observations*
While the accuracy is high, the nature of the responses suggests a search for the most probable/optimized path. There is little to no variation in the logical framework or the mathematical style.

## 4. Diversity Scoring
The diversity score is based on the variety of mathematical models, the range of variables prioritized, and the inclusion of non-standard heuristics.

*Table 21.* Comparison of Problem-Solving Diversity

| Feature | File 1 | File 2 |
|---|---|---|
| Modeling Variety | High (Fractional, Linear, Polynomial) | Low (Standardized Fractional) |
| Heuristic Diversity | Included (Trial & Error, Substitution) | Absent (Pure Algebraic) |
| Unit Handling | Diverse (Minutes, Hours, Mixed) | Uniform (Decimal Hours) |
| Logical Paths | Multiple (Solve $s$ first, solve $t$ first) | Single (Convergent on $s$) |
| **Diversity Score** | **9.5 / 10** | **3.0 / 10** |

## 5. Conclusion

**File 1** is the superior file in terms of diversity. It explores the problem through various mathematical lenses, ranging from standard algebra to human-like trial and error. **File 2**, while correct and efficient, is repetitive and lacks the breadth of thought required for "novel and diverse" output.

