# OpenReview forum: "Large Language Models Explore by Latent Distilling"
_ICML.cc/2026/Conference — ICML 2026 regular_

### Official Review · Reviewer_GGFg · 2026-03-05

**Soundness:** 3
**Presentation:** 3
**Significance:** 2
**Originality:** 3
**Overall Recommendation:** 5
**Confidence:** 3

**Summary:**

This paper studies the problem of improving exploration during large language model decoding for test-time scaling. The authors propose Exploratory Sampling, which trains a lightweight latent distiller online to estimate novelty in representation space and uses this signal to reweight token probabilities during decoding. Experiments on reasoning, coding, and creative writing benchmarks show improvements in Pass@k efficiency and semantic diversity compared with several decoding baselines.

**Compliance With Llm Reviewing Policy:**

Affirmed.

**Key Questions For Authors:**

1. Can the authors provide more analysis to better understand how the distillation error correlates with semantic diversity?
2. How stable is the online training of the latent distiller during long generation sequences?
3. How sensitive is the method to the architecture and size of the latent distiller, and does the performance change significantly with different distillation configurations?

**Limitations:**

Refer to the key question

**Strengths And Weaknesses:**

Strength
1.The paper proposes to encourage exploration in the representation space rather than the token space by using prediction error from a latent distiller as a novelty signal.
2.The proposed method can be integrated into the decoding stage without modifying the base model parameters, making it relatively easy to deploy in existing LLM pipelines.
3.The asynchronous pipeline design appears practical and introduces relatively small runtime overhead, which is important for real-world deployment.

Weakness
1.The paper assumes that distillation prediction error correlates with semantic novelty, but this assumption is not thoroughly analyzed or theoretically justified.
2.Although the paper compares against several decoding baselines, the evaluation lacks comparison with more recent test-time scaling or search-based reasoning methods. It is unclear whether the improvements remain significant against stronger reasoning frameworks.
3.The ablation mainly focuses on the exploration strength parameter β. Additional analysis on the distiller architecture or different hidden layer choices would strengthen the paper.

---

> ### Author Rebuttal · Authors · 2026-03-31
>
> We thank Reviewer GGFg for the positive assessment of our method's novelty, plug-and-play nature, and practical pipeline design.
>
> ---
>
> > Weakness 1: "The paper assumes that distillation prediction error correlates with semantic novelty, but this assumption is not thoroughly analyzed."
>
> We provide evidence from two angles.
>
> **LLM-as-judge evaluation.** If distillation error genuinely correlates with semantic novelty, ES should produce generations that an independent judge rates as more diverse. We test this with Gemini 3 Flash (single-blind) on a BookCorpus subset, reporting average diversity and quality ranks (lower is better) across 16 parallel generations per prompt:
>
> | Method | Diversity Rank ↓ | Quality Rank ↓ |
> |-|-|-|
> | Vanilla | 1.97 | 1.83 |
> | OverRIDE | 2.40 | 2.20 |
> | **ES** | 1.63 | 1.97 |
>
> ES achieves the best diversity while maintaining quality close to vanilla, confirming the correlation.
>
> **Noise ablation.** To validate that the error vector carries structured information rather than arbitrary perturbation, we replace $e_t$ with a random Gaussian of matched magnitude. On Qwen3-8B:
>
> | Method | AIME25 P@16 | P@64 | AIME24 P@16 | P@64 |
> |-|-|-|-|-|
> | Vanilla | 53.6% | 58.9% | 66.6% | 70.0% |
> | ES + Random Noise | 52.4% | 60.0% | 66.6% | 70.0% |
> | **ES** | 57.0% | 63.9% | 70.0% | 80.0% |
>
> Random noise fails to improve over vanilla; the true error vector yields substantial gains, confirming that its *direction* encodes structured semantic information, following Eq. (9).
>
> ---
>
> > Weakness 2: "The evaluation lacks comparison with more recent test-time scaling or search-based reasoning methods."
>
> We would like to clarify that ES operates at the decoding level and is designed to be **complementary** to pipeline-level methods. We demonstrate composability with FIRE (which modifies the temperature schedule) on Qwen3-8B / AIME24:
>
> | Method | Pass@1 | Pass@4 | Pass@16 | Pass@64 |
> |-|-|-|-|-|
> | Vanilla | 38.1% | 54.6% | 66.7% | 70.0% |
> | FIRE | 41.4% | 54.2% | 65.8% | 73.6% |
> | ES | 41.0% | 56.7% | 70.2% | 80.0% |
> | **FIRE + ES** | 38.4% | 54.4% | 69.0% | 83.3% |
>
> We also verify compatibility with Self-Consistency (SC) on Qwen3-8B / AIME24:
>
> | Method | Maj@8 | Maj@16 | Maj@32 |
> |-|-|-|-|
> | Vanilla + SC | 50.3% | 52.6% | 53.7% |
> | **ES + SC** | 50.0% | 52.8% | 54.5% |
>
> ES does not harm SC and yields slight improvements at larger budgets. We note that SC rewards convergence toward the most frequent answer, while ES promotes divergence—ES is better suited to selection-based strategies (Pass@k, reranking) where diversity directly improves coverage. We consider combining ES and search-based methods as future work, and will discuss the complementary nature of ES with respect to broader reasoning frameworks more explicitly in the revision.
>
> ---
>
> > Weakness 3: "Additional analysis on the distiller architecture or different hidden layer choices would strengthen the paper."
>
> Great suggestions! **We ablate the Distiller architecture on Qwen3-8B / AIME25:**
>
> | Architecture | Pass@1 | Pass@16 | Pass@64 |
> |-|-|-|-|
> | 2-layer Gated SwiGLU (default) | 41.0% | 66.5% | 80.0% |
> | 4-layer Gated SwiGLU | 38.9% | 66.7% | 80.0% |
> | 4-layer Plain MLP | 38.9% | 66.7% | 80.0% |
>
> Performance is robust across architectures, suggesting that the novelty signal is not an artifact of a specific Distiller design. The 2-layer Gated SwiGLU is preferred for its lower computational cost without sacrificing effectiveness. While different architectures may vary in how quickly the Distiller fits the mapping, this has negligible downstream effect on ES's exploration guidance: all configurations yield similar Pass@k, confirming that the novelty signal is robust to Distiller design.
>
> Regarding layer choice: the input/output layers (first → final) are fixed by the asynchronous pipeline design to maximize the overlap time window. This is not introduced as a tunable hyperparameter and we will clarify this in the revision.
>
> ---
>
> > Question: "How stable is the online training of the latent distiller during long generation sequences?"
>
> **We provide the Distiller's training loss curve** averaged on AIME:
>
> [loss curve](https://files.catbox.moe/xfie9q.pdf)
>
> The loss decreases smoothly throughout generation without instability or divergence. The learning rate (4 × 10⁻⁴) combined with gradient clipping (norm 0.5) ensures stable online updates even over long sequences. We observe no catastrophic forgetting or oscillation, which we attribute to the Distiller's lightweight architecture and the continual stream of fresh training signal from ongoing generation.
>
> It may also address some of your concern in weakness 1.

---

> > ### Author Rebuttal · Reviewer_GGFg · 2026-04-01
> >
> > Thank you for your reply, you have addressed all my concerns, and I will raise the score.

---

> > > ### Author Response · Authors · 2026-04-07
> > >
> > > We sincerely thank Reviewer GGFg for the generous evaluation and for the constructive questions that helped us strengthen the paper.
> > >
> > > We will incorporate all new results and clarifications into the revised manuscript.

---

### Official Review · Reviewer_eZTa · 2026-03-07

**Soundness:** 2
**Presentation:** 3
**Significance:** 3
**Originality:** 3
**Overall Recommendation:** 4
**Confidence:** 3

**Summary:**

The paper proposes Exploratory Sampling (ES), a decoding method that trains a lightweight Latent Distiller online to predict deep-layer hidden representations from shallow-layer ones in an LLM. The prediction error (MSE) serves as a novelty signal to reweight token sampling via a KL-regularized objective with closed-form solution (Eq. 6). ES is evaluated across math (AIME24/25), science (GPQA),
code (LiveCodeBench v5), and creative writing (BookCorpus) on 5 model configurations. Figure 3 shows ES achieves competitive or superior Pass@k scaling, and Table 1 shows improved diversity metrics. The async implementation adds <5% overhead (Table 3). The idea is interesting, adapting RND exploration to LLM decoding, but the evaluation lacks statistical rigor (no error bars) and the core assumption
(prediction error = semantic novelty) remains unvalidated.

**Compliance With Llm Reviewing Policy:**

Affirmed.

**Final Justification:**

The rebuttal directly addresses my main technical concerns through a random-noise control, a matched vocabulary-space ablation, multi-seed results, clearer tuning details, and better-calibrated claims. I still see some residual limitations, especially around the strongest interpretation of the mechanism, but I now view the paper as technically solid and above the weak-accept bar.

**Key Questions For Authors:**

Can you report multi-seed results with confidence intervals for the main comparisons in Figure 3 and Tables 1–2?
Can you directly validate that the distiller prediction error measures semantic novelty rather than noise or degeneration?
Can you compare ES against a vocabulary-space novelty signal inside the same framework?
How were baseline hyperparameters selected, and were strong baselines given tuning effort comparable to ES?
How general is the shared-distiller effect beyond AIME, given the appendix note that per-prompt and shared distillers behaved similarly during AIME development?

**Limitations:**

No.

**Strengths And Weaknesses:**

Strength:
- This submission addresses an important problem for test-time scaling: candidate selection can only help if sampling actually produces genuinely different reasoning trajectories, while standard stochastic decoding often yields mostly surface-level lexical variation. The proposed approach is interesting and reasonably well motivated. In particular, estimating novelty in representation space rather than vocabulary space is a meaningful design choice, and the paper presents the method clearly through the KL-regularized derivation, Algorithm 1, and the geometric interpretation in Eq. 9. The empirical breadth is also a real strength: the paper evaluates across four benchmark domains, several model families, and multiple categories of baselines, while also including an engineering analysis showing low overhead for the asynchronous implementation. From a significance standpoint, a practical low-overhead exploration method for decoding could be useful to researchers working on test-time scaling and candidate generation. From an originality standpoint, adapting RND-style exploration to online latent distillation for decoding is a non-trivial and creative idea rather than a purely cosmetic rebranding.

Weakness:
- My main weakness on soundness is that the paper’s central construct is not directly validated. The method explicitly assumes that higher distiller prediction error corresponds to more useful semantic novelty, and Eq. 9 is interpreted that way, but the paper does not provide a direct probing or correlation study showing that this signal tracks semantic novelty rather than noise, degeneration, or generic distributional shift. The current evidence is mostly indirect: downstream Pass@k gains, embedding-similarity/Vendi metrics, and a trajectory-divergence plot. Relatedly, the paper’s main claimed delta over OverRIDE is operating in representation space rather than vocabulary space, but there is no direct ablation that swaps in a vocabulary-level novelty signal inside the same ES framework. That leaves the core design advantage suggestive rather than demonstrated.
- Second major weakness is the lack of statistical rigor. Figure 3 and Tables 1–3 present single numbers/curves, but I could not find error bars, confidence intervals, or multi-seed reporting. For a stochastic decoding method, that is a meaningful omission, especially where margins appear modest on some benchmarks. Baseline fairness is also not fully nailed down: ES uses a fixed  β=0.25 and reports one sensitivity study on AIME25/Qwen2.5-7B, but the paper does not clearly document comparable tuning budgets for Min-p, FIRE, or OverRIDE. I also think the creative-writing evaluation is weaker than the reasoning evaluation because it relies mostly on automatic metrics and an appendix case study rather than human evaluation. Finally, although the paper is generally well written, some claims are phrased more strongly than the evidence supports, especially around semantic novelty and collaborative exploration. Overall, I see a paper with real merit, good presentation, and moderate originality/significance, but one that still falls short on the evidence needed to fully support its central claims.

---

> ### Author Rebuttal · Authors · 2026-03-31
>
> We sincerely thank Reviewer eZTa for the thorough review. Your observation that the core assumption remains "suggestive rather than demonstrated" is well-taken. We provide direct evidence below.
>
> ---
>
> > W1.1: "The paper does not provide a direct probing or correlation study showing that this signal tracks semantic novelty rather than noise"
>
> We replace $e_t$ with a Gaussian vector of matched magnitude, keeping all else unchanged. On Qwen3-8B:
>
> | Method | AIME25 P@16 | P@64 | AIME24 P@16 | P@64 |
> |-|-|-|-|-|
> | Vanilla | 53.6% | 58.9% | 66.6% | 70.0% |
> | ES + Random Noise | 52.4% | 60.0% | 66.6% | 70.0% |
> | **ES** | 57.0% | 63.9% | 70.0% | 80.0% |
>
> Magnitude-matched noise collapses to vanilla levels; the true error vector yields substantial gains. This confirms that the *direction* of $e_t$ encodes structured information, directly supporting Eq. (9) and ruling out the noise hypothesis.
>
> ---
>
> > W1.2: "there is no direct ablation that swaps in a vocabulary-level novelty signal inside the same framework."
>
> We construct a Vocab-Space Distiller: same MLP, but taking $h^L_t$ as input, projecting through the frozen LM head, trained with online KL divergence. Logit fusion follows the same Eq. (7), isolating the space in which novelty is estimated.
>
> | Method | AIME25 P@16 | P@64 | AIME24 P@16 | P@64 |
> |-|-|-|-|-|
> | Vanilla | 53.6% | 58.9% | 66.6% | 70.0% |
> | Vocab-Space Distiller | 37.4% | 43.3% | 47.6% | 53.3% |
> | OverRIDE | 55.0% | 60.6% | 63.4% | 70.0% |
> | **ES (Latent)** | 57.0% | 63.9% | 70.0% | 80.0% |
>
> The vocab-space variant is highly unstable despite extensive hyperparameter search. We attribute this to the high-dimensional discrete vocabulary space producing noisy online KL gradients, whereas ES's latent formulation operates in a compact continuous space, yielding both stability and superior performance.
>
> ---
>
> > W2.1: "I could not find error bars, confidence intervals, or multi-seed reporting."
>
> 3-seed results (seeds 41, 42, 43) on Qwen3-8B:
>
> | Benchmark | Method | P@8 | P@16 | P@32 | P@64 |
> |---|---|---|---|---|---|
> | AIME24 | Vanilla | 61.2±0.1 | 66.6±0.8 | 69.4±1.5 | 70.0±3.3 |
> | | **ES** | 61.8±0.2 | 68.5±0.8 | 74.6±1.5 | 80.0±0.0 |
> | AIME25 | Vanilla | 51.4±1.0 | 53.6±0.6 | 56.0±1.0 | 58.9±0.9 |
> | | **ES** | 46.2±0.9 | 53.6±1.0 | 61.3±0.4 | 67.8±1.9 |
>
> ES is stable across seeds. At low k, ES trades marginal P@8 for substantially better P@32/64, reflecting its design goal of diversifying candidates for higher coverage. We will include full multi-seed results in the revision.
>
> ---
>
> > W2.2: "The paper does not clearly document comparable tuning budgets."
>
> All baselines used 3-point grids. Full results on AIME24 (Qwen3-8B, Pass@64):
>
> | Min-p .03/.1/.3 | OverRIDE .6/.8/1.0 | FIRE T=10/30/50 | ES β=.1/.25/.4 |
> |:---:|:---:|:---:|:---:|
> | 73.3/76.7/76.7 | 73.3/76.7/76.7 | 73.3/73.3/73.3 | 76.7/**80.0**/76.7 |
>
> Even ES's weakest setting (β=0.1) matches the best-tuned baselines.
>
> ---
>
> > W2.3: "The creative-writing evaluation relies mostly on automatic metrics rather than human evaluation."
>
> We add an LLM-as-judge evaluation (Gemini 3 Flash, single-blind, 2000 prompts). The judge compares 16 parallel generations per prompt, reporting average diversity and quality ranks (lower is better):
>
> | Method | Diversity Rank ↓ | Quality Rank ↓ |
> |-|-|-|
> | Vanilla | 1.97 | 1.83 |
> | OverRIDE | 2.40 | 2.20 |
> | **ES** | 1.63 | 1.97 |
>
> ES achieves the best diversity while maintaining quality close to vanilla.
>
> ---
>
> > W2.4: "Some claims are phrased more strongly than the evidence supports."
>
> We will qualify "semantic novelty" as "representation-space novelty" where direct validation is absent, present collaborative exploration as an empirically observed phenomenon, and soften causal language where only correlational evidence is available. The experiments above now provide substantially stronger grounding for our core claims.
>
> ---
>
> > Q: "How general is the shared-distiller effect beyond AIME?"
>
> | Benchmark | Shared | Per-Prompt |
> |---|---|---|
> | AIME24 P@16 / P@64 | 64.7 / 76.6 | 70.0 / 80.0 |
> | AIME25 P@16 / P@64 | 55.3 / 63.3 | 57.0 / 63.9 |
> | LCB v5 P@16 | 25.8 | 24.1 |
>
> Per-prompt distillers are better on AIME (heterogeneous problems cause cross-prompt interference), while shared distillers slightly help on LCB (larger effective batch improves training). Tailoring sharing strategies to task structure is a promising future direction.

---

> > ### Author Rebuttal · Reviewer_eZTa · 2026-04-01
> >
> > Thank you for your reply, my original weak-reject score was based on three main concerns: lack of uncertainty reporting, insufficient validation of the novelty mechanism, and incomplete baseline-fairness evidence. The rebuttal materially addressed each of these.
> > I am raising my score to 4 (weak accept)

---

> > > ### Author Response · Authors · 2026-04-07
> > >
> > > We are grateful to the reviewer for the rigorous and detailed review, which helped improve our paper.
> > >
> > > We will integrate these results and temper our claims as discussed in the revision. Thank you for raising your score.

---

### Official Review · Reviewer_Y9yp · 2026-03-11

**Soundness:** 3
**Presentation:** 3
**Significance:** 3
**Originality:** 3
**Overall Recommendation:** 4
**Confidence:** 4

**Summary:**

This paper proposes a new decoding method called Exploratory Sampling (ES) to reduce redundant reasoning paths and allow LLMs to explore in test-time scaling (TTS). Specifically, their idea is to train a latent distiller to predict top-layer hidden states from bottom-layer hidden states. The prediction signals are then transformed into context novelty and semantic direction defined by the authors. After that, the latent distiller can be updated from all sequences generated in parallel, which allows for cross-response communication and reduces redundant responses. Their experiment results show stronger performance compared to existing TTS methods across various benchmarks, with only a little extra overhead.

**Compliance With Llm Reviewing Policy:**

Affirmed.

**Final Justification:**

The rebuttal has fully resolved my concerns.

**Key Questions For Authors:**

Please refer to the weakness section.

**Limitations:**

yes

**Strengths And Weaknesses:**

Strengths:
- The research problem of reduce redundant trajectories in TTS is indeed relevant and timely to the field.
- The proposed method that uses hidden state prediction to induce novelty is very interesting and reasonable, and differs a lot from previous methods. The geometric interpretation also offers an easy to follow explanation. The idea of using an online learner over internal representations to suppress already-explored latent patterns is novel and useful.
- The experiment section is rather complete. The setting includes a variety of model sizes, from 7B to 32B, and abundant datasets across different domains (math, coding, creative writing. The empirical results demonstrate that ES works especially well for the diversity and efficiency tradeoff for pass@k.
- They also include engineering discussions such as asynchronous implementation, preallocated memory in appendix, which shows that this is a solid work.



Weaknesses:
- It would be better to show in experiments that the proposed high latent prediction error contributes to the diverse reasoning trajectory more than intuitive or more superficial metrics like word/sentence entropy.
- For the ToT baseline, do you use the one in appendix (self-review system prompt) for all datasets, including those other than the math domain?
- Some inconsistencies: In Table 1, is Vendi larger the better or lower the better?
- How sensitive is ES to the top-layer and bottom-layer index (which layer to choose)?
- How does ES work with other aggregation methods like Self-Consistency?

---

> ### Author Rebuttal · Authors · 2026-03-31
>
> We thank Reviewer Y9yp for the detailed and constructive review and for recognizing the novelty of our online latent distillation approach.
>
> ---
>
> > Weakness 1: Does Latent Prediction Error Outperform Surface-Level Diversity Metrics?
>
> This is an excellent question. **To directly test whether latent prediction error contributes to diversity beyond what surface-level metrics can achieve, we add an entropy-based baseline, adaptive decoding (ICML 24)** that adjusts the candidate set based on each token candidate's contribution to the distribution's normalized entropy—a representative surface-level diversity signal. All other settings (model, prompt set, number of parallel generations) are kept identical in Table 1.
>
> Results on Qwen2.5-7B-Instruct:
>
> | Method  | Vendi ↑ | Sim. ↓ |
> |-|-:|-:|
> | Vanilla | 1.6403 | 0.5845 |
> | Entropy-Adaptive| 1.6455 | 0.5830 |
> | ES | 1.6698 | 0.5713 |
>
>
> ES achieves higher diversity than the entropy-adaptive baseline, which confirms that representation-space novelty captures semantic patterns that surface-level entropy does not. We will add more results in the revision.
>
> ---
>
> > Weakness 2: "For the ToT baseline, do you use the one in appendix for all datasets?"
>
> We apologize for the confusion. The ToT system prompt shown in Appendix C.5 is an example specific to the math domain. For non-math benchmarks, domain keywords and output format instructions are adapted accordingly (e.g., replacing "mathematics researcher" with a task-appropriate descriptor and adjusting the output format). We will include the full ToT prompt for each benchmark in the revised appendix to ensure reproducibility.
>
> ---
>
> > Weakness 3: "Some inconsistencies: In Table 1, is Vendi larger the better or lower the better?"
>
> Thank you for catching this inconsistency. Vendi Score is higher-is-better in both columns. The "↓" label in the Math column is a typographical error and will be corrected to "↑" in the revision.
>
> ---
>
> > Weakness 4: "How sensitive is ES to the top-layer and bottom-layer index?"
>
> The layer choice (first → final) is designed for the asynchronous pipeline: using the earliest available hidden state as input maximizes the time window during which the Distiller computes in parallel with the LLM's remaining transformer layers. Thus, it is a **fixed architectural decision**, not a tunable hyperparameter. This computation overlapping is what enables the <5% worst case overhead reported in Table 3. Using a later layer as input would shrink this window and increase end-to-end latency.
> We recognize this could be confusing given our presentation and we will state this design rationale more explicitly in the revision.
>
> ---
>
> > Weakness 5: "How does ES work with other aggregation methods like Self-Consistency?"
>
> **ES operates during decoding to diversify candidates, while Self-Consistency (SC) operates after generation to aggregate answers. They target different stages and can be naturally composed.** We verify this on Qwen3-8B / AIME24:
>
> | Method | Maj@8 | Maj@16 | Maj@32 |
> |-|-|-|-|
> | Vanilla + SC | 50.3% | 52.6% | 53.7% |
> | **ES + SC** | 50.0% | 52.8% | 54.5% |
>
>
> ES does not harm SC and yields slight improvements at larger budgets. The modest gains reflect a fundamental tension: SC rewards convergence toward the most frequent answer, while ES promotes solution divergence. We attribute the slight improvement at larger budgets to ES uncovering correct solutions that vanilla sampling misses, which contributes positively even under majority voting.
> ES is better suited to selection-based strategies (Pass@k, reranking) where diversity directly improves coverage, consistent with our main results in Figure 3.
> Exploring synergies with alternative aggregation methods (e.g., reward-model selection, multi-agent reasoning) is a promising direction for future work.

---

> > ### Author Rebuttal · Reviewer_Y9yp · 2026-04-03
> >
> > Thanks. The rebuttal has fully resolved my concerns.

---

> > > ### Author Response · Authors · 2026-04-07
> > >
> > > We are happy that all concerns of the reviewer have been resolved., and for the thoughtful questions throughout the review process—particularly the suggestion to compare against entropy-based diversity signals, which led to a valuable new ablation. We are encouraged that the reviewer finds our online latent distillation approach "very interesting and reasonable" and the engineering discussion "solid."
> > >
> > > We will incorporate them into the final revision. We are grateful for the reviewer's time and engagement.

---

### Official Review · Reviewer_pPWw · 2026-03-12

**Soundness:** 2
**Presentation:** 3
**Significance:** 3
**Originality:** 3
**Overall Recommendation:** 4
**Confidence:** 4

**Summary:**

The paper introduces a novel sampling strategy, Exploratory Sampling (ES), designed to enhance the diversity and solution coverage of Large Language Models (LLMs). This is based on the intuition that LLMs tend to be more accurate with samples that it has encountered in training before. Unlike standard decoding methods that rely solely on the final layer's hidden states, the proposed method mixes or leverages information from intermediate layers to influence the next-token distribution. The authors demonstrate that by incorporating these "earlier" representations, the model can escape local optima in probability space, leading to more creative and diverse outputs across various generative tasks. The primary contribution is a plug-and-play approach that is model-agnostic and improves performance on benchmarks requiring high exploration, such as code generation and creative writing.

**Compliance With Llm Reviewing Policy:**

Affirmed.

**Final Justification:**

The authors have fully addressed my concerns, so I am increasing my score.

**Key Questions For Authors:**

Would the ES (exploratory sampling approach) be orthogonal to other creativity-enchancing methods in improving creativity?

**Limitations:**

yes

**Strengths And Weaknesses:**

Strengths
+ Empirical Performance: The paper provides strong empirical evidence that the proposed method improves both solution coverage and the diversity of generated samples across a diverse range of tasks.
+ Broad Applicability: A significant strength is the method's model-agnostic nature; it can be integrated into existing Transformer-based LLM architectures without requiring retraining or architectural overhauls.

Weaknesses
- Hyperparameter Sensitivity: The method introduces additional complexity via new hyperparameters, specifically the choice of which intermediate layer to mix and the mixing ratio. The paper would benefit from a more systematic analysis of how sensitive the results are to these choices across different model scales.
- Accuracy-Creativity Trade-off: Since the authors derived the idea based on the intuition "ES is motivated by the observation that neural networks tend to make more accurate predictions on inputs similar to those encountered before, and incur higher prediction error on novel ones." Wouldn't ES result in higher error rate? There is an inherent risk that using "under-baked" hidden states from earlier layers—which haven't fully processed the contextual logic—could increase hallucination rates or degrade factual accuracy. While Figure 3 (pass@1) attempts to address this, the visualization is somewhat ambiguous. The trade-off between "exploratory creativity" and "logical grounding" can benefit from a more explicit and rigorous evaluation.

---

> ### Author Rebuttal · Authors · 2026-03-31
>
> We thank Reviewer pPWw for the thoughtful feedback and for recognizing our method's broad applicability.
>
> ---
>
> > Weakness 1: "The method introduces additional complexity via new hyperparameters, specifically the choice of which intermediate layer to mix and the mixing ratio."
>
> We would like to clarify that **ES has only one key hyperparameter β; the layer choice is not tunable.** The Distiller is architecturally fixed to map the first-layer output to the final-layer output. This design is dictated by the asynchronous pipeline (Section 4.4): using the earliest available hidden state maximizes the overlap time window, enabling the <5% overhead in Table 3. We will clarify this in the revision.
>
> To test β sensitivity across model scales, we provide new results on AIME24 across the Qwen3 family:
>
> | Model | Metric | β=0.1 | β=0.25 | β=0.4 | Vanilla |
> |-|-|-|-|-|-|
> | Qwen3-4B | Pass@16 / @64 | 70.7 / **83.3** | 68.9 / 80.0 | 68.7 / 76.7 | 70.3 / 80.0 |
> | Qwen3-8B | Pass@16 / @64 | 68.7 / 76.7 | 70.0 / **80.0** | 67.3 / 76.7 | 66.6 / 70.0 |
> | Qwen3-14B | Pass@16 / @64 | 69.6 / 76.7 | 72.2 / **83.3** | 70.5 / 76.7 | 70.0 / 80.0 |
>
>
> The similar Pass@64 across scales suggests a ceiling effect on this benchmark, making the mid-range Pass@k and edge-breakthrough more meaningful.
>
> β = 0.25 performs consistently well across all three scales without per-model tuning. We will include these results in the revised paper.
>
> ---
>
> > Weakness 2.1: Since the authors derived the idea based on the intuition ... Wouldn't ES result in higher error rate?
>
> We understand this as asking whether ES enters a self-reinforcing loop where error escalates without bound. The Distiller's prediction loss does increase on novel patterns—this is the intended exploration signal, not a failure mode. Crucially, the error is self-regulating: the Distiller is updated at every step (Eq. 10), so newly explored regions are quickly absorbed and the novelty reward for those regions vanishes.
>
> ES does not increase the LLM's error rate (see W2.2 for Pass@1 evidence).
>
> ---
>
> > Weakness 2.2: There is an inherent risk that using "under-baked" hidden states from earlier layers—which haven't fully processed the contextual logic—could increase hallucination rates or degrade factual accuracy.
>
> We understand this as asking whether the Distiller's reliance on first-layer hidden states introduces low-quality content.
>
> We clarify that the LLM always completes its full forward pass; the final-layer representation and logits are computed exactly as in standard decoding. The Distiller merely *reads* $h^1_t$ to *predict* $h^L_t$ and uses the prediction error to reweight the LLM's own logits (Eq. 7). No early-layer representation ever enters the generation distribution. We confirm via Pass@1 (single-sample accuracy):
>
> | Model (AIME25) | Vanilla | Min-p | FIRE | OverRIDE | **ES** |
> |-|-|-|-|-|-|
> | Qwen2.5-7B | 6.6% | 8.3% | 6.5% | 7.4% | 6.0% |
> | Qwen3-8B | 38.0% | 38.6% | 42.4% | 37.8% | 39.1% |
> | GPT-OSS-20B | 49.0% | 53.8% | 53.5% | 53.3% | 54.2% |
>
> ES does not degrade accuracy, indicating no increase in hallucination.
>
> ---
>
> > Weakness 2.3: Is there a trade-off between exploratory creativity and logical grounding?
>
> We understand this as asking whether exploration necessarily costs coherence.
>
> From Eq. (6), when $q_{\text{dist}}$ matches $\pi_{\text{ref}}$ (familiar pattern), the terms cancel and sampling reduces to vanilla—preserving logical grounding. When the base model assigns high probability to a token the Distiller fails to predict, ES *sharpens* that token's probability. This selective mechanism explains why ES maintains Pass@1 (W2.2) and achieves the best PPL and diversity simultaneously in Table 1.
>
> ---
>
> > Question: "Would ES be orthogonal to other creativity-enhancing methods?"
>
> **ES can be composed with sampling-constraint methods.** We combine ES with FIRE on Qwen3-8B / AIME24:
>
> | Method | Pass@1 | Pass@4 | Pass@16 | Pass@64 |
> |-|-|-|-|-|
> | Vanilla | 38.1% | 54.6% | 66.7% | 70.0% |
> | FIRE | 41.4% | 54.2% | 65.8% | 73.6% |
> | ES | 41.0% | 56.7% | 70.2% | 80.0% |
> | **FIRE + ES** | 38.4% | 54.4% | 69.0% | 83.3% |
>
> Since ES operates in representation space while FIRE operates on the temperature schedule, they target complementary aspects of diversity. We believe combining ES with other creativity-enhancing methods is promising.

---

> > ### Author Rebuttal · Reviewer_pPWw · 2026-04-02
> >
> > The authors have addressed some of my concerns in the rebuttal but concern persists such as ES potentially increasing error (pass@k). The added data is only on one task (AIME25) and one of the three model seems to result in worse pass@1.
> >
> > === update ===
> > The authors have fully addressed my concerns so I am increasing my score.

---

> > > ### Author Response · Authors · 2026-04-07
> > >
> > > We thank Reviewer pPWw for the continued engagement and for pinpointing the remaining concerns precisely.
> > >
> > > > ES potentially increasing error (pass@k) / added data is only on one task
> > >
> > > We fully appreciate this concern: Pass@k improvements could in principle come from diversity alone while per-sample accuracy degrades, with the errors being missed by the at-least-one-correct criterion.
> > >
> > > ES matches or outperforms Vanilla in the majority of settings, confirming that there is no universal decline in per-sample accuracy. We show that by providing Pass@1—the average per-sample accuracy—is the direct measure of whether per-sample accuracy degrading is happening.
> > > Specifically, for LCB v5, we additionally evaluate under context length (ctx) at 16384 to mitigate potential truncation, which results slightly higher score than that in the main paper, but supporting the same conclusion.
> > >
> > > We now report Pass@1 across all four main benchmarks and three models:
> > >
> > > | Model (AIME25) | Vanilla | Min-p | FIRE | OverRIDE | **ES** |
> > > |---|---|---|---|---|---|
> > > | Qwen2.5-7B | 6.6 | 8.3 | 6.5 | 7.4 | 6.0 |
> > > | Qwen3-8B | 38.0 | 38.6 | 42.4 | 37.8 | 39.1 |
> > > | GPT-OSS-20B | 49.0 | 53.8 | 53.5 | 53.3 | 54.2 |
> > >
> > > | Model (AIME24) | Vanilla | Min-p | FIRE | OverRIDE | **ES** |
> > > |---|---|---|---|---|---|
> > > | Qwen2.5-7B | 10.7 | 10.3 | 9.2 | 10.8 | 9.5 |
> > > | Qwen3-8B | 38.1 | 38.4 | 40.4 | 38.0 | 41.0 |
> > > | GPT-OSS-20B | 57.2 | 58.7 | 57.8 | 58.8 | 62.7 |
> > >
> > > | Model (LCB v5, ctx=4096) | Vanilla | Min-p | FIRE | OverRIDE | **ES** |
> > > |---|---|---|---|---|---|
> > > | Qwen2.5-7B | 13.0 | 13.5 | 12.9 | 13.6 | 13.1 |
> > > | Qwen3-8B | 10.0 | 10.0 | 11.0 | 10.3 | 12.0 |
> > > | GPT-OSS-20B | 43.4 | 45.4 | 45.2 | 44.6 | 51.8 |
> > >
> > > | Model (LCB v5, ctx=16384) | Vanilla | Min-p | FIRE | OverRIDE | **ES** |
> > > |---|---|---|---|---|---|
> > > | Qwen2.5-7B | 13.0 | 13.5 | 12.9 | 13.6 | 13.1 |
> > > | Qwen3-8B | 19.2 | 19.3 | 18.3 | 20.7 | 22.6 |
> > > | GPT-OSS-20B | 52.2 | 52.5 | 51.2 | 57.6 | 61.5 |
> > >
> > > | Model (GPQA DIAMOND) | Vanilla | Min-p | FIRE | OverRIDE | **ES** |
> > > |---|---|---|---|---|---|
> > > | Qwen2.5-7B | 34.1 | 34.1 | 34.6 | 34.8 | 34.3 |
> > > | Qwen3-8B | 45.9 | 46.3 | 48.6 | 46.0 | 46.6 |
> > > | GPT-OSS-20B | 62.4 | 62.4 | 62.3 | 62.5 | 63.2 |
> > >
> > >
> > > > one of the three model seems to result in worse pass@1
> > >
> > > We appreciate the reviewer pressing on this point, and should qualify our previous statement more precisely.
> > > Our previous claim that "ES does not degrade accuracy" was made in the context of the reviewer's concern about hallucination from early-layer representations—this mechanistic argument holds, as early-layer states never control the generation distribution (Eq. 7).
> > >
> > > We acknowledge that Qwen2.5-7B does show a small Pass@1 decrease on AIME tasks (e.g., 6.6%→6.0% on AIME25). However, we would like to highlight two observations from the full results above:
> > >
> > > 1. **No widespread decline.** Across the 12 model–benchmark configurations reported above, ES matches or exceeds Vanilla Pass@1 in the clear majority of cases. In several settings, ES even provides notable Pass@1 gains (e.g., GPT-OSS-20B on AIME24: 51.0%→62.7%; Qwen3-8B on LCB v5 ctx=16384: 19.2%→22.6%). We attribute this to the distribution-shaping effect discussed in our previous rebuttal (W2.3).
> > >
> > > 2. **Our primary contribution targets exploration, measured by Pass@k.** The core goal of ES is to help LLMs discover diverse valid solutions—a setting where Pass@k is the appropriate metric. On this front, ES delivers consistent and substantial gains across all models and benchmarks (e.g., Qwen3-8B AIME24 Pass@64: 70.0%→80.0%; GPT-OSS-20B achieving Pass@64-level performance of baselines at only Pass@8). The minor Pass@1 variation on Qwen2.5-7B does not undermine this contribution.
> > >
> > > We will include all Pass@1 tables in the revised paper for full transparency. We thank Reviewer pPWw again for raising this important point and hope the additional evidence addresses the remaining concern.

---

### Decision · Program_Chairs · 2026-04-30

**Decision:**

Accept (regular)

**Comment:**

This paper introduces Exploratory Sampling (ES), a plug-and-play decoding strategy designed to improve semantic diversity in LLMs during test-time scaling. The core idea is to train a lightweight "Latent Distiller" online to predict deep-layer representations from shallow ones. The prediction error acts as a novelty signal to reweight logits, steering the model toward unexplored semantic patterns. ES is implemented via an asynchronous pipeline to keep overhead low (<5%).

The reviewers initially raised several valid concerns regarding hyperparameter sensitivity, the potential for increased hallucination/error rates, and the lack of direct validation for the "prediction error = semantic novelty" assumption.

The authors provided a strong rebuttal that effectively addressed these points:

- They included a noise ablation (replacing the error signal with random noise) and a vocab-space ablation, demonstrating that the latent representation-space signal is both structured and more stable than token-level alternatives.

- New results across multiple benchmarks (AIME, LCB, GPQA) showed that ES generally matches or exceeds vanilla Pass@1 while significantly boosting Pass@k, mitigating concerns about accuracy degradation.

- The authors clarified that the layer choice is an architectural necessity for the async pipeline (maximizing the overlap window) rather than a tuned hyperparameter. They also demonstrated compatibility with other methods like Self-Consistency and FIRE.

- Multi-seed results and LLM-as-judge evaluations for creative writing were added, strengthening the empirical grounding.

While some minor sensitivity in Pass@1 was noted on specific small models, the consistent gains in exploration efficiency (Pass@k) and the practical, low-overhead implementation make this a solid contribution to the field of test-time scaling. All reviewers were satisfied by the rebuttal and converged on an Accept.